# Breast cancer plasticity is restricted by a LATS1-NCOR1 repressive axis

Yael Aylon [1,10] ✉, Noa Furth [2,10], Giuseppe Mallel [1], Gilgi Friedlander [3], Nishanth Belugali Nataraj [2], Meng Dong [4], Ori Hassin [1], Rawan Zoabi[1], Benjamin Cohen[5], Vanessa Drendel[6], Tomer Meir Salame[7], Saptaparna Mukherjee[1], Nofar Harpaz[2], Randy Johnson[8], Walter E. Aulitzky[9], Yosef Yarden [2], Efrat Shema [2] & Moshe Oren [1] ✉

Breast cancer, the most frequent cancer in women, is generally classified into several distinct histological and molecular subtypes. However, single-cell technologies have revealed remarkable cellular and functional heterogeneity across subtypes and even within individual breast tumors. Much of this heterogeneity is attributable to dynamic alterations in the epigenetic landscape of the cancer cells, which promote phenotypic plasticity. Such plasticity, including transition from luminal to basal-like cell identity, can promote disease aggressiveness. We now report that the tumor suppressor LATS1, whose expression is often downregulated in human breast cancer, helps maintain luminal breast cancer cell identity by reducing the chromatin accessibility of genes that are characteristic of a "basal-like" state, preventing their spurious activation. This is achieved via interaction of LATS1 with the NCOR1 nuclear corepressor and recruitment of HDAC1, driving histone H3K27 deacetylation near NCOR1-repressed "basal-like" genes. Consequently, decreased expression of LATS1 elevates the expression of such genes and facilitates slippage towards a more basal-like phenotypic identity. We propose that by enforcing rigorous silencing of repressed genes, the LATS1-NCOR1 axis maintains luminal cell identity and restricts breast cancer progression.

Lineage plasticity, the switching of cells from one morphological and functional identity to another, has been recognized as a cardinal property essential for embryonic development and tissue homeostasis[1]. This highly regulated process goes awry when tumors exploit this inherent plasticity to their own advantage. For instance, cell plasticity can generate intratumoral diversity that exacerbates tumor progression, partially by the emergence of populations with metastatic potential. Mounting evidence suggests that epigenetic mechanisms, including aberrant histone modifications and dysfunction of chromatin remodelers, play key roles in promoting lineage plasticity in cancer[2–5] and phenotypic diversification of breast cancer[6,7].

[1]Department of Molecular Cell Biology, The Weizmann Institute of Science, 76100 Rehovot, Israel. [2]Department of Immunology and Regenerative Biology, The Weizmann Institute of Science, 76100 Rehovot, Israel. [3]Department of Life Sciences Core Facilities, The Nancy & Stephen Grand Israel National Center for Personalized Medicine (G-INCPM), The Weizmann Institute of Science, 76100 Rehovot, Israel. [4]Dr. Margarete Fischer-Bosch-Institute of Clinical Pharmacology and University of Tuebingen, Stuttgart, Germany. [5]Department of Immunology, The Weizmann Institute of Science, 76100 Rehovot, Israel. [6]Department of Pathology, Robert Bosch Hospital, Stuttgart, Germany. [7]Flow Cytometry Unit, Department of Life Sciences Core Facilities, The Weizmann Institute of Science, 76100 Rehovot, Israel. [8]Department of Cancer Biology, University of Texas MD Anderson Cancer Center, Houston, TX 77030, USA. [9]Department of Hematology, Oncology and Palliative Medicine, Robert Bosch Hospital, Stuttgart, Germany. [10]These authors contributed equally: Yael Aylon, Noa Furth. ✉e-mail: yael.aylon@weizmann.ac.il; moshe.oren@weizmann.ac.il

Breast cancer is a heterogeneous disease encompassing different histological and molecular subtypes, with distinct clinical behaviors[8–12]. Two-thirds of all breast cancers express estrogen receptor (ER) and are classified as luminal A (lumA) or luminal B (lumB), with lumB tumors being more proliferative and heterogeneous[13,14]. In luminal tumors, ER is the driving transcription factor, whose target genes control proliferation and endocrine response[15]. An additional 15–20% of breast cancers are basal-like and predominantly triple-negative for hormone receptors, thus limiting targeted therapeutic options and giving rise to tumors associated with poor overall survival and high relapse rates[16,17]. Although the biological basis of subtype distinction remains poorly understood, experimental studies have demonstrated that basal-like tumors originate from a luminal progenitor cell population[18,19].

The acquisition of basal-like markers in luminal tumors is associated with metastasis[20], often occurring in the pro-metastatic invasive front of human luminal breast tumors[21,22] and is correlated with treatment failure[23,24]. Together, these observations illustrate the hazard of luminal tumors that harbor a basal-like cell component. Importantly, mutations in tumor suppressors can shift luminal progenitors towards a more basal-like state[18,25–29].

Transcriptional corepressors, such as Nuclear COre Repressive complex 1 and 2 (NCOR1 and NCOR2), are crucial regulators of ligand-induced ERα-mediated phenotypes, actively repressing ERα-downregulated genes[30–37]. NCOR1 repressive activity and ability to decrease chromatin accessibility are regulated by its ability to recruit histone deacetylases (HDACs), including HDAC3 and HDAC1[38–41]. NCOR function is further regulated by post-translational modifications and nuclear-cytoplasmic shuttling[42–51].

Interestingly, expression of both ERα and NCOR is downregulated during progression from intraductal to invasive ductal breast carcinoma[52]. Moreover, driver mutations in ERα and NCOR are mutually exclusive in breast cancer metastasis[53,54]. Of note, inactivating mutations in either gene arise with a markedly increased frequency in breast cancer patients treated with anti-endocrine therapy[53,55], reflecting their function in mediating the repressive activity of ERα and tamoxifen sensitivity[56]. Together, these observations indicate that the inactivation of the repressive function of ERα can contribute to phenotypic evolution and facilitate the emergence of more aggressive and therapy-resistant cancers.

In recent years, the Hippo pathway kinases LArge Tumor Suppressors 1 and 2 (LATS1 and LATS2, respectively) have become the focus of intense research[57]. LATS1/2 expression is downregulated in breast cancer[58–63] and is associated with aggressive phenotypes such as increased tumor size, lymph node metastases, and poor prognosis[61,62,64]. The early view of LATS1 and LATS2 as redundant paralogs, which function solely within the Hippo pathway to phosphorylate and inactivate the transcriptional cofactors YAP and TAZ[65,66], has recently been expanded. In particular, it has been reported that the LATS proteins can act in a YAP/TAZ-independent manner to restrict ERα activity[67]. Intriguingly, others have reported that Hippo pathway proteins actually induce ERα activity[68–72]. Elucidating distinct functions of LATS1 and LATS2 might help resolve these apparent discrepancies; for instance, we previously reported that depletion of LATS1 but not LATS2 in the mouse MMTV-PyMT model, which shares features with human lumB breast cancer, results in the adoption of more basal-like characteristics and increased resistance to tamoxifen[61].

Here we report the use of single-cell profiling to generate a high throughput multi-parametric analysis of cellular epigenetic heterogeneity in mammary tumors. Specifically, we describe the existence of an epigenetically distinct basal-like population within luminal tumors. LATS1 thwarts this luminal-to-basal-like phenotypic cell plasticity by augmenting NCOR1 repressive activity and facilitating H3K27ac deacetylation. We propose that by ensuring the shutoff of ERα-repressed genes, the LATS1–NCOR1 axis maintains luminal cell identity, thereby restricting the progression of luminal breast cancer.

## Results

Polyomavirus middle T oncogene (PyMT)-driven mammary tumors (MMTV-PyMT) are a commonly used mouse model of breast cancer[73–75]. Cancer progression and gene expression patterns in this model[76] share common features with human lumB tumors[77]. Therefore, it is often used as a proxy for lumB cancer. It is noteworthy that MMTV-PyMT tumors show progressive loss of ER expression, which is observable also in a subset of lumB tumors but usually not in lumA tumors[78,79].

Using a custom-designed antibody panel for Epigenetic-focused CyTOF (EpiTOF)[80,81] (see Table S1), we performed high-dimensional phenotypic classification of single cells from five PyMT tumors from 3.5-month-old female mice. Including known markers of cell differentiation and identity[82–86] in the CyTOF panel enabled to assign of the tumor epithelial cells to subpopulations representing different phenotypic states: luminal progenitors ("lumP"), mature luminal ("luminal"), and a small but discrete subpopulation displaying molecular features that are commonly associated with a basal-like state ("basal-like") (Fig S1a), as well as nonepithelial tumor-associated cells such as fibroblasts and immune cells. This is consistent with the notion that human luminal tumors often include basal-like cell subpopulations[82,83,87–92]. Interestingly, the EpiTOF panel revealed that these subpopulations differ epigenetically; the two modifications that most consistently distinguished between the mature luminal and basal-like cell populations were H3K36me2 (higher in luminal) and H3K27ac (higher in basal-like) (Fig S1b, c), reflecting the critical role of epigenetic modifications in defining cell identity[84].

Next, we derived primary cell cultures from PyMT tumors and subjected them to EpiTOF analysis. Reassuringly, the in vitro cultured cells recapitulated the in vivo phenotypic distribution (Fig. 1a, left panel). Remarkably, depletion of the *Lats1* tumor suppressor (Lats1-CKO) increased the fraction of basal-like cells at the expense of mature luminal cells in vitro (Fig. 1a, right panel, and Fig S1d), in concordance with our earlier observation that Lats1-CKO mammary tumors are enriched for basal-like features[61]. Intriguingly, the Lats1-CKO cultures were also enriched, to various degrees, in a subpopulation of cells defined as fibroblasts, based on their expression of LATS1 (deleted in the epithelial compartment) and CD44[93] (Figs. 1a and S1d, right). Echoing the higher levels of H3K27ac in basal-like cells and H3K36me2 in luminal cells (Fig. S1b,c), these epigenetic marks were also the most differential between the WT and Lats1-CKO cultures (Fig. 1b). Specifically, H3K27ac tended to be higher in Lats1-CKO cells, whereas H3K36me2 was higher in WT cells, in all three subpopulations: luminal progenitor (lumP), luminal and basal-like (Fig. 1c). The global differences in H3K27ac and H3K36me2 levels between WT and Lats1-CKO cells, confirmed by Western blots (Fig. 1d), suggest that LATS1 may maintain luminal cell identity, at least in part, by stabilizing defined epigenetic states.

Expression of epithelial cell adhesion molecule (EpCAM) and Integrin α6 (CD49f) is commonly used to distinguish between luminal and basal-like mammary cells[25,91,94–99]. Luminal progenitor cells are defined as EpCAM+/CD49f+[94,99–101] and are thought to be the precursors to both luminal and basal-like tumors[91]. To further investigate the impact of LATS1 on cell identity, we, therefore, utilized these cell surface markers, which strongly segregated between the luminal/luminal progenitor (henceforth called "luminal") and basal-like subpopulations of both WT and Lats1-CKO PyMT cancer cells (Fig. 1e).

Next, WT and Lats1-CKO cultures were FACS-separated into defined luminal or basal-like subpopulations (Fig. 2a), briefly expanded, and subjected to global gene expression (RNA-seq) (Fig S2a) and chromatin accessibility (ATAC-seq) analyses. To validate the cell identity of the FACS-sorted subpopulations, we compared the expression levels (Fig S2b), chromatin accessibility (Fig S2c), and staining intensity (Fig S2d) of conventional luminal (Krt8 and Krt18) and basal (Krt14 and Krt5) markers. Of note, these markers may be

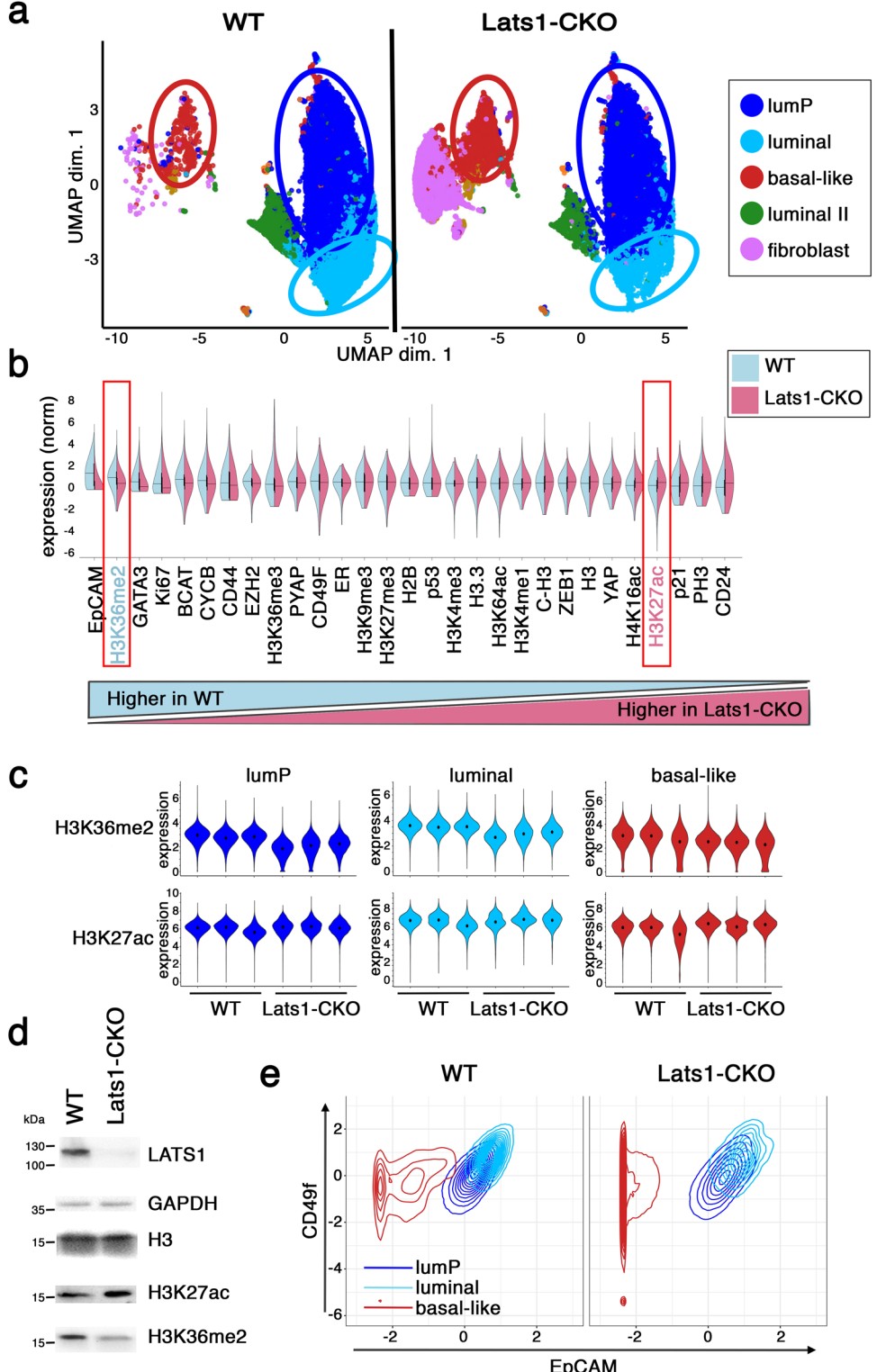

**Fig. 1 | *Lats1* knockout promotes an epigenetically distinct basal-like population. a** Single cells derived from 3 WT (left panel) and 3 littermate-matched Lats1-CKO (right panel) PyMT-driven tumors were processed for EpiTOF analysis as described in the "Methods" section. Shown is a uniform manifold approximation and projection for dimension reduction (UMAP). Relevant populations are indicated by circles. LumP luminal progenitors. **b** Comparison of the relative levels of EpiTOF markers in the WT vs. Lats1-CKO EpiTOF-defined in vitro basal-like subpopulations. Expression of each of the markers was centered to zero mean and displayed as a split violin plot ordered by directionality and extent of the difference.

Black lines designate medians. Red boxes denote H3K36me2 and H3K27ac. **c** Violin plot comparing mean expression levels of the indicated epigenetic marks in WT vs. Lats1-CKO PyMT cells in the three different subpopulations of the in vitro EpiTOF samples (defined in **a**). **d** Western blot analysis of H3K27ac and H3K36me2 in WT and Lats1-CKO in vitro samples. GAPDH served as a loading control. Total H3 levels are also shown. Representative blot of five biological repeats. **e** Bivariate contour plots showing the expression (Z-scores) of EpCAM and CD49f in WT vs. Lats1-CKO in vitro EpiTOF samples, within the indicated subpopulations.

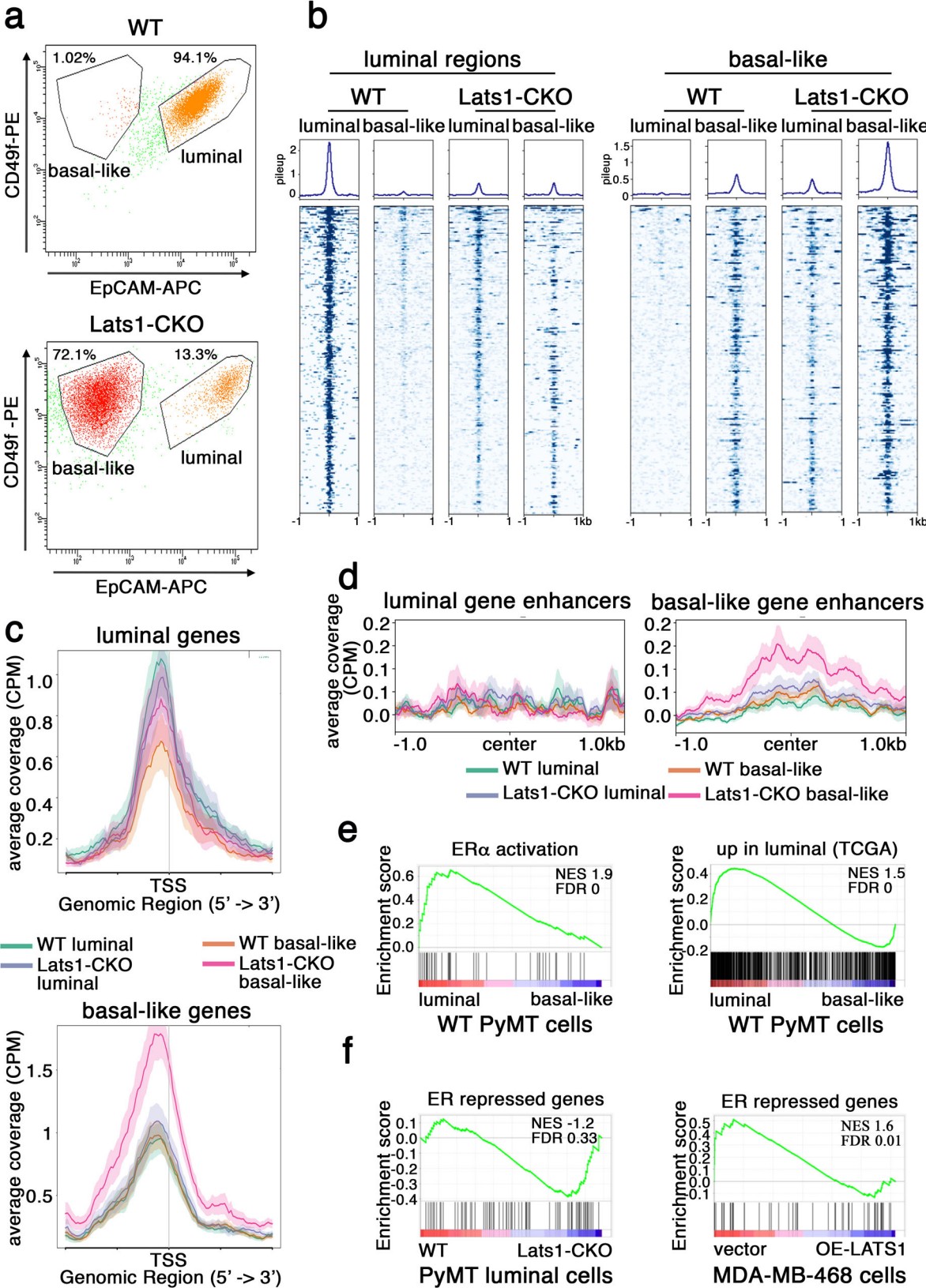

expressed to some extent also in tumors that are defined as luminal[102], supporting the notion that tumors display greater heterogeneity than normal breast tissues and therefore actually constitute a spectrum of phenotypic states rather than a discrete luminal or basal-like state. As expected, luminal markers displayed more "open" chromatin and were more highly expressed in the luminal subpopulations. However, cells

lacking LATS1 displayed a drift towards a more basal-like state; even when sorted as luminal by the surface abundance of EpCAM and CD49f, they showed chromatin and keratin staining patterns partly resembling basal-like cells (Fig. S2c, d).

The ATAC-seq analysis of the different FACS-sorted subpopulations showed that overall, the total numbers of ATAC-seq peaks were

**Fig. 2 | Characterization of luminal and basal-like subpopulations. a** WT and Lats1-CKO PyMT-derived cell lines were FACS-sorted according to relative EpCAM and CD49f expression. Solid black lines demark luminal and basal-like sub-populations isolated for brief expansion in culture. **b** WT and Lats1-CKO FACS-enriched luminal and basal-like subpopulations were propagated to attain a minimum of 50,000 cells for analysis by ATAC-seq. Signals associated with WT luminal peaks (266 peaks, left) or WT basal-like peaks (212 peaks, right) are presented. Two biological replicates were analyzed to call differential peaks; one representative sample from each condition is depicted. **c** Cumulative ATAC-seq reads over TSS of genes differentially upregulated (RNA-seq analysis using *DESeq2* FC > 1.5, raw *p*-value < 0.05) in luminal (top) or basal-like (bottom) cells, in the indicated sub-populations. Two biological replicates were analyzed and the resultant ATAC-seq BAM files were merged. Lines depict average coverage, shaded areas represent SE. **d** Comparison of cumulative ATAC-seq reads over enhancers associated with genes differentially upregulated (RNA-seq analysis using *DESeq2* FC > 1.5, raw *p*-value < 0.05) in luminal (left, 92 enhancers) or basal-like (right, 374 enhancers) cells in WT and Lats1-CKO PyMT cells. Genes were associated with enhancers based on ENC+ EDP enhc-Gene dataset[166]. Merged coverage from replicates is shown. Lines depict average coverage, shaded areas represent SE. **e** Gene set enrichment analysis (GSEA) of WT luminal vs. WT basal-like differential gene expression compared with an ERα activated[30] gene set (left) and with genes upregulated in human luminal tumors relative to basal-like tumors (TCGA) (right). **f** GSEA of PyMT WT vs. Lats1-CKO luminal (left) and MDA-MB-468 cells expressing vector control or LATS1 after 48 h of doxycycline induction (right), compared to an ER repressed[30] gene set.

similar between the Lats1-CKO and WT subpopulations (Table S2). However, distinct genotype and phenotype-specific patterns of distribution of chromatin accessibility were clearly discernable. In particular, WT luminal-specific regions that were inaccessible in WT basal-like cells (Fig. 2b, left, WT), tended to be less prominent but equally open in Lats1-CKO luminal and Lats1-CKO basal-like cells (Fig. 2b, left, Lats1-CKO). Furthermore, WT "basal" regions, defined as selectively open in WT basal-like but not WT luminal cells, were more open in Lats1-CKO basal-like cells (Fig. 2b, right). Importantly, while practically inaccessible in WT luminal cells, these "basal" peaks became markedly accessible in Lats1-deficient luminal cells (Fig. 2b, right). Extending the observations in Fig. S2c, this ATAC-seq data further confirmed that Lats1-deficient luminal cells, although expressing luminal surface markers, retain "basal" regions in an open chromatin conformation. Expectedly, chromatin openness (deduced from ATAC-seq) was positively correlated with gene expression (deduced from RNA-seq) (Fig S2e).

Collectively, these data imply that whereas WT cells maintain defined luminal and basal-like cell states, with distinct chromatin architectures, cells lacking *Lats1* are unable to restrict chromatin accessibility in cell identity-defining regions. This may allow higher phenotypic plasticity and reduced barriers to transition from luminal to a more basal-like state, explaining the increased abundance of cells with basal-like features in Lats1-CKO tumors and cultures.

Next, chromatin accessibility associated with genes expressed differentially between luminal and basal-like WT cells (defined by our RNA-seq), was compared between the four subpopulations (WT luminal; WT basal-like; Lats1-CKO luminal and Lats1-CKO basal-like). As expected, WT luminal cells maintained the chromatin of "luminal" genes most open, while WT basal-like cells kept these genes in a closed chromatin state (Fig. 2c, top). In contrast, Lats1-CKO cells maintained a moderately open chromatin state of these genes, irrespective of whether they expressed luminal or basal-like cell surface markers (Fig. 2c, top). Conversely, Lats1-CKO basal-like cells maintained the chromatin of "basal" genes in a highly open state, but the WT cells retained these genes relatively closed, even within their basal-like subpopulation (Fig. 2c, bottom). Interestingly, enhancers associated with genes expressed preferentially in the basal-like state were more accessible in basal-like Lats1-CKO cells than in their WT counterparts (Fig. 2d, right), although "luminal" gene enhancers were not differentially accessible in any of the cell types (Fig. 2d, left). Together, these observations suggest that cells derived from PyMT mouse mammary carcinomas maintain a predetermined chromatin state depending on genotype: cells with intact LATS1 tend to retain a luminal chromatin setting, whereas cells lacking LATS1 are predisposed to adopt a basal-like chromatin architecture, thereby reducing epigenetic barriers and enabling them to slip more readily into a more basal-like phenotypic state.

As expected, estrogen receptor signaling ("ERα activation"[30]) was significantly enriched among the genes differentially expressed in WT luminal, compared to WT basal-like cells (Fig. 2e, left and Fig S2f). Concordantly, along with other motifs (Table S3), estrogen response elements (ERE) were significantly enriched in our ATAC-seq "luminal"

peaks (Fig. 2b, left, and Table S4). Importantly, the gene expression pattern of our EpCAM/CD49f-defined luminal subpopulation was significantly similar to a human luminal breast cancer gene signature (Fig. 2e, right), further supporting the human relevance of the MMTV-PyMT model.

Remarkably, compared to their WT luminal counterparts, Lats1-CKO luminal cells displayed elevated expression of genes defined in two independent datasets as repressed by ERα ("ER repressed genes"[30] Fig. 2f, left and "ER repressed genes"[103] Fig. S2g). Expression of such genes was suppressed also upon overexpression of LATS1 in MDA-MB-468 human basal-like breast cancer cells (Fig. 2f, right). Notably, similar to "basal" gene enhancers, enhancers of genes repressed by ER were more "open" in Lats1-CKO luminal cells, compared to WT luminal cells (Fig. S2h). Correspondingly, ERE motifs were significantly over-represented also in chromatin regions accessible only in basal-like state (Fig. 2b, right and Table S4). These differences were not accompanied by changes in the amount of ERα protein expression (Fig. S2i).

Together, this supports the conjecture that in both PyMT mouse mammary carcinoma cells and human breast cancer cell lines, LATS1 may maintain luminal identity by facilitating the repression of a subset of genes, including ERα-repressed genes, which should remain silent in luminal cells. In line with this notion, genes more highly expressed in Lats1-CKO luminal cells than in WT luminal cells were consistently associated with chromatin regions more accessible ("up") in basal-like (BL) than in luminal state (Fig. S2j).

To further explore the plasticity of tumor cell state, the basal-like subpopulation was FACS-sorted from WT PyMT tumor-derived cells (Fig. 3a, panel 2) and its phenotypic composition was followed over time. Remarkably, within six passages of the enriched basal-like cells, a significant portion of the population is already presented as luminal (Fig. 3a, panel 3). We then re-purified the basal-like population of these cells by FACS sorting (Fig. 3a, panel 4); once more, within seven additional passages, approximately half the population was already presented again as luminal (Fig. 3a, panel 5). Moreover, with additional passaging this "mixed" cell culture shifted gradually further towards an even more luminal state (Fig. 3a, panels 6, 7). Interestingly, as cells became less basal-like, genes upregulated upon *Lats1* deletion (Fig. S2j) were less transcribed (Fig. 3b). In contrast, the basal-like cell sub-population that was FACS-sorted from Lats1-CKO cells, maintained its basal-like phenotype over multiple passages (Fig. S3a). Together, this demonstrates that PyMT mouse mammary carcinoma cells that retain LATS1 expression have an inherent preference to maintain a stable luminal cell state.

To assess cancer cell plasticity in vivo, FACS-separated luminal and basal-like subpopulations from WT and Lats1-CKO cell lines were injected into mammary fat pads of syngeneic mice, and tumors were analyzed 4 weeks later. Remarkably, whereas WT luminal cells retained their luminal identity within the tumors, ~80% of the Lats1-CKO luminal cells transitioned to a basal-like phenotype (Fig. 3c). Concordantly, these tumors displayed markedly different histologies; whereas WT luminal cells generated poorly differentiated carcinomas, Lats1-CKO luminal cells generated basal-like sarcomatoid carcinomas (Fig. S3b).

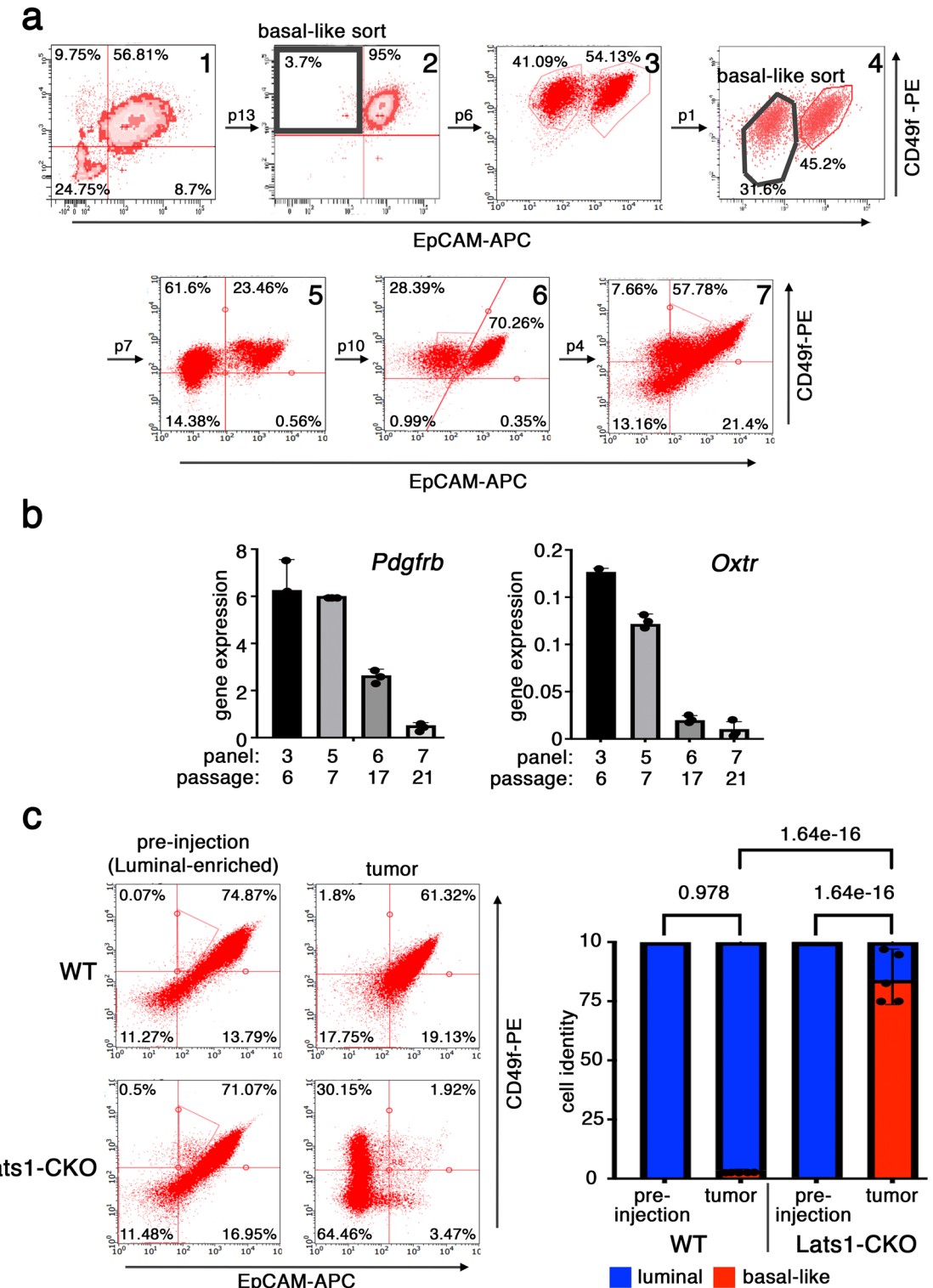

Furthermore, within the timeframe of these experiments, only the Lats1-CKO cells gave rise to lung metastases (Fig. S3b). Intriguingly, the injection of purified basal-like cells of either genotype did not yield any palpable tumors (Fig. S3c), suggesting that this subpopulation may lack tumor-initiating capacity. However, when luminal cells were injected together with genetically matched basal-like cells, this combination yielded larger tumors than the same number of luminal cells injected alone (Fig. S3c). Thus, luminal-to-basal-like transitioned cells appear to augment tumor aggressiveness. Together, our data suggest that LATS1 may impose a barrier to bolster luminal cell identity and

restrict cancer cell plasticity also in vivo, which may restrain tumor growth and curb its metastatic potential.

To implicate LATS1 more directly in the prevention of luminal-to-basal-like transition, we re-expressed an MYC-tagged version of mouse LATS1 in three independently derived Lats1-CKO cell lines (Fig. S4a). These cells tolerated only limited LATS1 expression, presumably reflecting an anti-proliferative phenotype of excessive LATS1. Despite this low expression, in each case, reconstitution of LATS1 resulted in a marked increase in the portion of cells displaying a luminal phenotype (Fig. 4a).

**Fig. 3 | Luminal and basal-like subpopulations display phenotypic plasticity.**
**a** Dissociated cells from a freshly harvested WT PyMT tumor were profiled by FACS (top, panel 1), using antibodies for EpCAM and CD49f. Basal-like cells are defined as EpCAM$^{low}$CD49f$^{high}$ (upper left quadrant), while luminal cells (including also luminal progenitors) are defined as EpCAM$^{high}$CD49f$^{high}$ (upper right quadrant). After 13 passages in culture, the small basal-like subpopulation was enriched by FACS sorting (panel 2), followed by additional passaging in culture and FACS analysis (panel 3). The basal-like subpopulation was then enriched again (panel 4) and further cultured, with intermittent FACS profiling after the number of passages indicated above the arrows (panels 5–7). Bold black shapes labeled "basal-sort" depict the gates used for FACS-enrichment of the basal-like cell population. **b** WT PyMT cells were harvested in parallel to the FACS analyses, and mRNA was extracted and analyzed by RT-qPCR to quantify the expression levels of LATS1-repressed, "basal" genes. Values were normalized to *Hprt* expression. "panel" refers to the corresponding panel number in (**a**). The cumulative number of passages is also indicated. Data are presented as mean values ± SD of three technical replicates. Source data are provided as a Source Data file. **c** WT and Lats1-CKO luminal-enriched cells were injected into mouse mammary fat pads. Four weeks later, tumors were harvested, dissociated, and subjected to FACS analysis. Representative FACS profiles of cells pre-injection and from dissociated tumors are shown (left). A graphical representation of the relative portions of luminal (high EpCAM) and basal-like (low EpCAM) cells are shown on the right (mean ± SE of five mice from each group). One-way ANOVA was used to compute significance. Source data are provided as a Source Data file.

In agreement with the elevated H3K27ac levels in Lats1-CKO cells (Fig. 1d), LATS1 reconstitution elicited a decrease in global H3K27ac (Fig. 4b). Importantly, LATS1 expression was negatively associated with H3K27ac levels also in human breast-derived cells; knockout of *LATS1* (LATS1-KO) in non-transformed mammary epithelial MCF10A cells (Fig. 4c) or silencing of *LATS1* (siLATS1) in MCF7 luminal breast cancer cells (Fig. 4d) resulted in modest augmentation of H3K27ac levels. In contrast, induced overexpression of LATS1 in human basal-like breast cancer MDA-MB-468 cells diminished H3K27ac levels (Fig. 4e). Furthermore, depletion of LATS1 from luminal MCF7 or ZR751 cells resulted in a global enrichment of ER-repressed gene expression (Fig. 4f), in line with the notion that restraining H3K27ac may be associated with LATS1-dependent maintenance of the luminal identity of human breast cancer cells.

To assess the phenotypic impact of LATS1 in human breast cancer cells, we subjected MDA-MB-468 cells to FACS analysis, employing EpCAM as a luminal marker. As expected, MDA-MB-468 cells were largely EpCAM negative (Fig. 4f, middle), in line with their basal-like identity[104]. Importantly, even in these basal-like cells, the population displaying elevated LATS1 also gained EpCAM positivity (Fig. 4g, right). Remarkably, overexpression of LATS1 in MDA-MB-468 cells was sufficient to render their gene expression pattern more similar to that of luminal breast cancer tumors (TCGA-BRCA dataset) (Fig. 4h). Additionally, expression of the canonical luminal marker KRT8 was compared between LATS1 reconstituted Lats1-CKO cells and their parental Lats1-CKO controls. As seen in Fig. S4b, the restoration of LATS1 expression significantly augmented both the number of KRT8-expressing cells and the intensity of KRT8 expression. Thus, LATS1 can favor luminal cell identity in mouse and human breast cancer cells, in association with the downmodulation of H3K27ac.

YAP and TAZ, the two Hippo-pathway downstream effectors of LATS1 signaling, have been associated with basal-like cell identity[70,105–108]. To examine whether hyperactivation of YAP and/or TAZ upon depletion of LATS1 might drive the acquisition and maintenance of basal-like cell attributes in our experimental system, we established inducible knockdown of *Yap* or *Taz* in WT or Lats1-CKO basal-like-enriched cells (Fig. S4c); as expected, knockout of *Lats1* augmented, and *Yap* or *Taz* depletion decreased, the expression of *Cyr61*, a canonical YAP/TAZ target[109,110] (Fig. S4d). In line with our previous observations (Fig. 3a), WT basal-like cells acquired a mixed luminal/basal-like phenotype during culturing to attain stable knockdown pools, while Lats1-CKO cells retained a robust basal-like phenotype over multiple passages (shCont, Fig. S4e, f). However, contrary to expectations, following 14 days of YAP or TAZ depletion, WT basal-like cultures actually presented a moderately augmented, rather than attenuated, basal-like phenotype, while Lats1-CKO cultures (with initial high YAP/TAZ activity) fully retained their basal-like phenotype despite *Yap/Taz* depletion (Fig. S4e, f, shYAP and shTAZ). The phenotypic change in the YAP or TAZ-depleted WT basal-like cells was concurrent with more than a 50% decrease in expression of the luminal marker *Krt18* (Fig. S4g). Hence, in this experimental system, it seems strongly unlikely that LATS1-driven promotion of luminal state occurs via inhibition of YAP or TAZ.

To search for alternative mechanisms by which LATS1 might exert its pro-luminal effect, we queried our RNA-seq data for upstream regulators that might contribute to LATS1-dependent transcriptional differences. To enable more robust extrapolations, we performed Ingenuity Pathway Analysis[111] (IPA, QIAGEN Inc., https://digitalinsights.qiagen.com/IPA) on LATS1-dependent expression patterns from three different model systems: Lats1-CKO tumors[61], Lats1-CKO luminal-enriched cell lines, and depletion of LATS1 in MDA-MB-468 cells (each compared to its corresponding WT or vector control). Interestingly, this analysis suggested that NCOR1 activity was significantly compromised in all LATS1-depleted systems (Fig. 5a). NCOR1 is a well-characterized co-repressor of ERα[112,113] and its downregulation has been implicated in tamoxifen resistance[114–116], presumably due to loss of luminal cell identity. NCOR1 regulates chromatin accessibility by recruiting histone deacetylases, leading to H3K27 deacetylation, chromatin condensation, and subsequent repression of "basal" gene transcription[30,117].

In line with the notion that LATS1 might augment the ability of NCOR1 to act as a co-repressor, documented NCOR1-repressed genes (IPA database[111], Table S5) were indeed more highly expressed (Fig. 5b) and displayed more accessible chromatin (Fig. 5c) in Lats1-CKO cells, similarly to genes culled from Fig S2j (more highly expressed in Lats1-CKO cells and associated with more open chromatin in the basal-like state). Likewise, these genes were strongly repressed upon overexpression of LATS1 in MDA-MB-468 cells (Fig. 5d), while simultaneous silencing of NCOR1 and LATS1 (Fig S5a) maximized the expression of those genes (Fig. 5e). Concordantly, depletion of either NCOR1 or LATS1 diminished the expression of the luminal marker *Krt18* in WT cells, while increasing the expression of the basal-like marker *Krt14* (Fig S5b). Importantly, also in human luminal breast cancer MCF7 cells, depletion of either NCOR1 or LATS1 increased the levels of H3K27ac (Fig. 4d) and upregulated NCOR1-repressed genes (Fig S5c). Together, these results strongly suggest that LATS1 augments NCOR1-driven gene repression to promote luminal fate in human and mouse breast cancer cells.

Moreover, MYC-tagged mouse LATS1 co-immunoprecipitated with endogenous NCOR1 (Fig. 5f), and a specific, albeit weak, co-precipitation of endogenous LATS1 with endogenous NCOR1 was also observable in WT cells (Fig. S5d), implying that both proteins are constituents of a common molecular complex. The endogenous interaction between LATS1 and NCOR1 was validated also in MCF7 cells (Fig. S5e). HDAC1, one of the histone deacetylases recruited by NCOR1 to mediate transcriptional repression[41,118], also associated with the ectopically expressed LATS1 (Fig. 5f), and in an endogenous setting was specifically co-immunoprecipitated with NCOR1 in cells expressing LATS1 but not in LATS1-deficient cells (Fig. S5d), suggesting that its recruitment to NCOR1 is LATS1-dependent.

The staining patterns of LATS1 and NCOR1 proteins were remarkably similar in PyMT mouse and MCF7 human mammary carcinoma cells (Figs. 5g and S5f). In both cases, although LATS1 was predominantly cytoplasmic, a small fraction appeared to be nuclear.

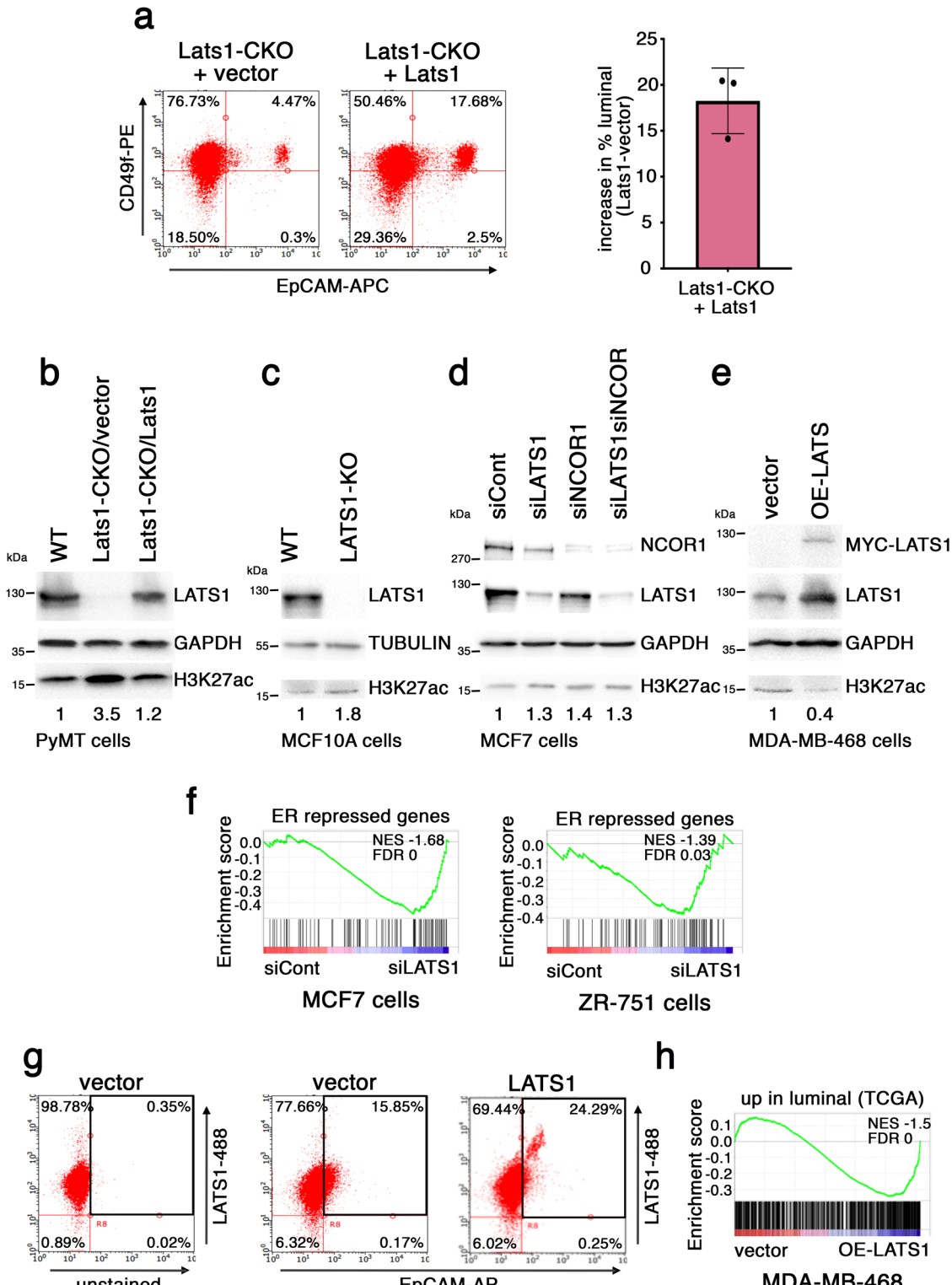

Additionally, NCOR1 nuclear abundance was augmented in cells overexpressing LATS1 (Fig. 5g, arrows). To assess more quantitatively the colocalization of NCOR1 and LATS1, and to exclude the possibility that the large GFP-tag might alter the subcellular distribution of LATS1, we performed ImageStream analysis using MYC-tagged mouse LATS1. This analysis revealed that about 15% of LATS1 (Fig. S5g, panel 1) and 93% of NCOR1 (Fig. S5g, panel 2) were nuclear (similar to DAPI staining). Despite the different subcellular localization of bulk LATS1 and NCOR1, remarkably, 80% of single cells revealed a significant overlap of LATS1 and NCOR1 distribution (Similarity > 1.5, Z-score = 0.013)

(Fig. S5g, panel 3). Moreover, in agreement with Fig. 5g, NCOR1 abundance was positively correlated with LATS1 expression; WT cells demonstrated stronger NCOR1 staining than Lats1-CKO cells, and LATS1 overexpression further augmented NCOR1 staining (p-value < 0.0001 in all comparisons) (Fig. S5g, panel 4). In line with these observations, the LATS1–NCOR1 interaction was predominantly (but not exclusively) nuclear, as detected by a proximity ligation assay (PLA) in both PyMT and MCF7 cells (Fig. S5h, i). Taken together, this suggests that LATS1 may promote the formation and/or stability of a nuclear repressive complex comprising LATS1, NCOR1, and HDAC1.

**Fig. 4 | LATS1 regulates phenotypic plasticity. a** MYC-tagged mouse *Lats1* or vector control was stably introduced into three independent Lats1-CKO cell lines. Left: representative FACS analysis as in Fig. 3a. Right: graphical representation of the proportional increase in luminal cells in each cell line upon LATS1 over-expression, measured as in (**a**, left) (mean ± SD of 3 cell lines). Source data are provided as a Source Data file. **b** Lats1-CKO PyMT cells stably harboring vector or MYC-tagged mouse *Lats1* were subjected to Western blot analysis with the indicated antibodies. A WT sample is presented in the first lane for comparison. GAPDH served as a loading control. Numbers under lanes represent relative H3K27ac band intensity, normalized to the corresponding loading control and control sample. Representative blot of five biological repeats. **c** MCF10A cells, either WT or with CRISPR/Cas9 deletion of LATS1 (LATS1-KO), were subjected to Western blot analysis with the indicated antibodies. Tubulin served as a loading control. Numbers are as in (**b**). Representative blot of two biological repeats. **d** MCF7 cells transiently transfected with the indicated siRNAs were subjected to Western blot analysis with the indicated antibodies. GAPDH served as a loading control. Numbers are as in (**b**). Representative blot of three biological repeats. **e** MDA-MB-468 cells harboring vector control or human MYC-LATS1, after 48 h of doxycycline induction, were subjected to Western blot analysis with the indicated antibodies. Top panel = 9E10 antibody, directed against the MYC-tag. GAPDH served as a loading control. Numbers are as in (**b**). Representative blot of four biological repeats. **f** GSEA of MCF7 (left) and ZR-751 (right) cells, transiently transfected with control siRNA (siCont) or LATS1 siRNA (siLATS1) (data from Furth et al.[61]), compared to an ER repressed[30] gene set. **g** FACS analysis of MDA-MB-468 cells harboring vector control or human MYC-LATS1, after 48 h of doxycycline induction. Cells were first stained for APC-EpCAM, and then permeabilized to stain intracellular LATS1 (probed with Alexa Fluor 488 dyed secondary antibody). Black square designates the quadrant of EpCAM^high luminal cells. **h** GSEA of MDA-MB-468 cells without (vector) vs. with induction of LATS1 (OE-LATS1), compared to genes upregulated in human luminal tumors relative to basal-like tumors (data from TCGA).

Next, we examined the impact of LATS1 on the selective recruitment of NCOR1 to chromatin. As expected, NCOR1 was strongly enriched on regulatory regions of NCOR1-repressed genes (Fig. 6a), but not on other genomic regions (Fig. S6a). Importantly, the depletion of LATS1 greatly reduced the target-specific association of NCOR1 with chromatin (Fig. 6a). Furthermore, in line with the notion that the LATS1–NCOR1–HDAC1 complex removes acetylation marks from histones adjacent to NCOR1-binding regions, H3K27ac levels at regulatory regions of NCOR1-repressed genes were inversely correlated with NCOR1 and LATS1 levels (Figs. 6b, S6b, left), while this was not the case for a genomic region not associated with NCOR1 (Fig S6b, right). Treatment with A-485, an inhibitor of the p300 histone acetyltransferase, dramatically decreased global H3K27ac (Fig. S6c) and abrogated the effect of *Lats1* depletion on the transcription of NCOR1-repressed genes (Figs. 5b and S2d), consistent with the conjecture that LATS1 restricts the expression of these genes by promoting histone deacetylation.

Interestingly, the pattern of distribution of H3K36me2, which was enriched in the luminal tumor subpopulation (Fig. S1b) and augmented in WT relative to Lats1-depleted cells (Fig. 1c, d), was opposite to that of H3K27ac. Specifically, the deletion of *Lats1* decreased H3K36me2 on LATS1–NCOR1-repressed regulatory regions (Fig. S6e). Of note, a negative association between H3K27ac and H3K36me2 has been observed also in other settings[119,120]. Importantly, upon shRNA-mediated depletion of NCOR1 for 2 weeks (Fig S6f), whereas Lats1-CKO cells were virtually unaffected, WT cells displayed a marked increase in the basal-like subpopulation, from about 3% to over 45% (Fig. 6c). Together, these data support a model in which LATS1–NCOR1–HDAC1-mediated transcriptional repression of a distinct set of genes contributes to the maintenance of luminal cell identity.

To assess the clinical implications of our findings, we analyzed available gene expression data from human breast tumors. Genes whose expression was significantly different in luminal B tumors possessing low levels of *NCOR1* (NCOR1^low) mRNA relative to high *NCOR1* mRNA expressers, or low levels of *LATS1* (LATS1^low) mRNA relative to high *LATS1* mRNA expressers, were examined (Fig. S7a). Strikingly, essentially all genes whose expression was differential in both NCOR1^low and LATS1^low tumors, exhibited a co-directional behavior: 461 and 228 genes were significantly co-downregulated or co-upregulated, respectively, in both LATS1^low and NCOR1^low tumors, whereas only two genes displayed opposite directionalities (Chi-squared test, *p*-value < 0.001). The 228 upregulated genes include also the human equivalents of our mouse LATS1–NCOR1–HDAC1 repressed genes (Fig. S7a) and presumably comprise additional targets of LATS1–NCOR1-mediated repression.

Of note, the expression of both *NCOR1* and *LATS1* was significantly lower in basal-like tumors, compared to luminal tumors (Fig. 7a); the differential expression of *LATS1* was already noted by us previously[61]. The lower expression of LATS1 and NCOR1 in basal-like cancers suggests that luminal tumors with diminished levels of either protein

might be more susceptible to adopting basal-like attributes. In line with this notion, ER-repressed genes appeared to be derepressed in NCOR1^low LATS1^low human luminal tumors, relative to their NCOR1^high LATS1^high counterparts (Fig. 7b). Remarkably, examination of a small set (*N* = 15) of luminal breast cancer patient samples showed a noticeable similarity of LATS1 and NCOR1 immunohistochemical staining patterns: tumors with nuclear LATS1 tended to have more nuclear NCOR1, whereas tumors with cytoplasmic LATS1 tended to harbor more cytoplasmic NCOR1 (Fig S7b). On the other hand, *LATS1* expression was significantly elevated in breast cancers harboring an inactivating mutation in the *NCOR1* gene, relative to those harboring wild-type *NCOR1* (Fig. 7c). This apparent mutual exclusivity between *NCOR1* mutations and *LATS1* downregulation suggests that NCOR1 inactivation and decreased LATS1 expression may have a redundant impact on breast cancer, in agreement with their proposed shared mechanism of action. Importantly, in ER-positive breast cancers, low *NCOR1* expression correlates with reduced relapse-free survival (RFS) (Fig. 7d, left). Interestingly, low *LATS1* expression is associated with even lower RFS in patients with NCOR1^low tumors (Fig. 7d, right); while this might seemingly be inconsistent with functional redundancy, one should keep in mind that although LATS1 and NCOR1 are relatively underexpressed in such tumors, none is fully depleted, leaving room for combined effects of their partial loss. Altogether, retention of proper LATS1 and NCOR1 activity appears to be beneficial for the survival of luminal breast cancer patients.

## Discussion

Our study exemplifies a paradigm of mechanistic transmission from decreased expression of tumor suppressor genes to consequent alterations in the epigenetic landscape and cell identity. Specifically, we report that LATS1 maintains luminal cell identity by associating with NCOR1 to secure the repression of genes that are silenced by estrogen receptors in luminal cells. This is accompanied by a defined chromatin state, ensuring that luminal genes remain "open" while genes that are associated with a basal-like state are kept inaccessible. Depletion or downregulation of LATS1, as occurs in many breast cancer tumors, leads to epigenetic promiscuity and predisposes breast cancer cells to express basal-like genes. Breast cancer cell plasticity caused by LATS1 dysfunction contributes to increased tumor aggressiveness and, presumably, therapy resistance.

Much of the work described above involved the use of cancer cells derived from MMTV-PyMT tumors. Although sharing many features with human lumB tumors, tumorigenesis in this model is driven by a viral oncoprotein (PyMT), which is not implicated in human breast cancer (or other human cancers). This obviously is a drawback when extrapolating findings from this mouse model to human breast cancer. Nevertheless, it should be noted that the PyMT viral oncoprotein promotes cancer by constitutive activation of the same pathways (SRC, RAS, PI3K) that are hyperactivated by breast cancer-relevant

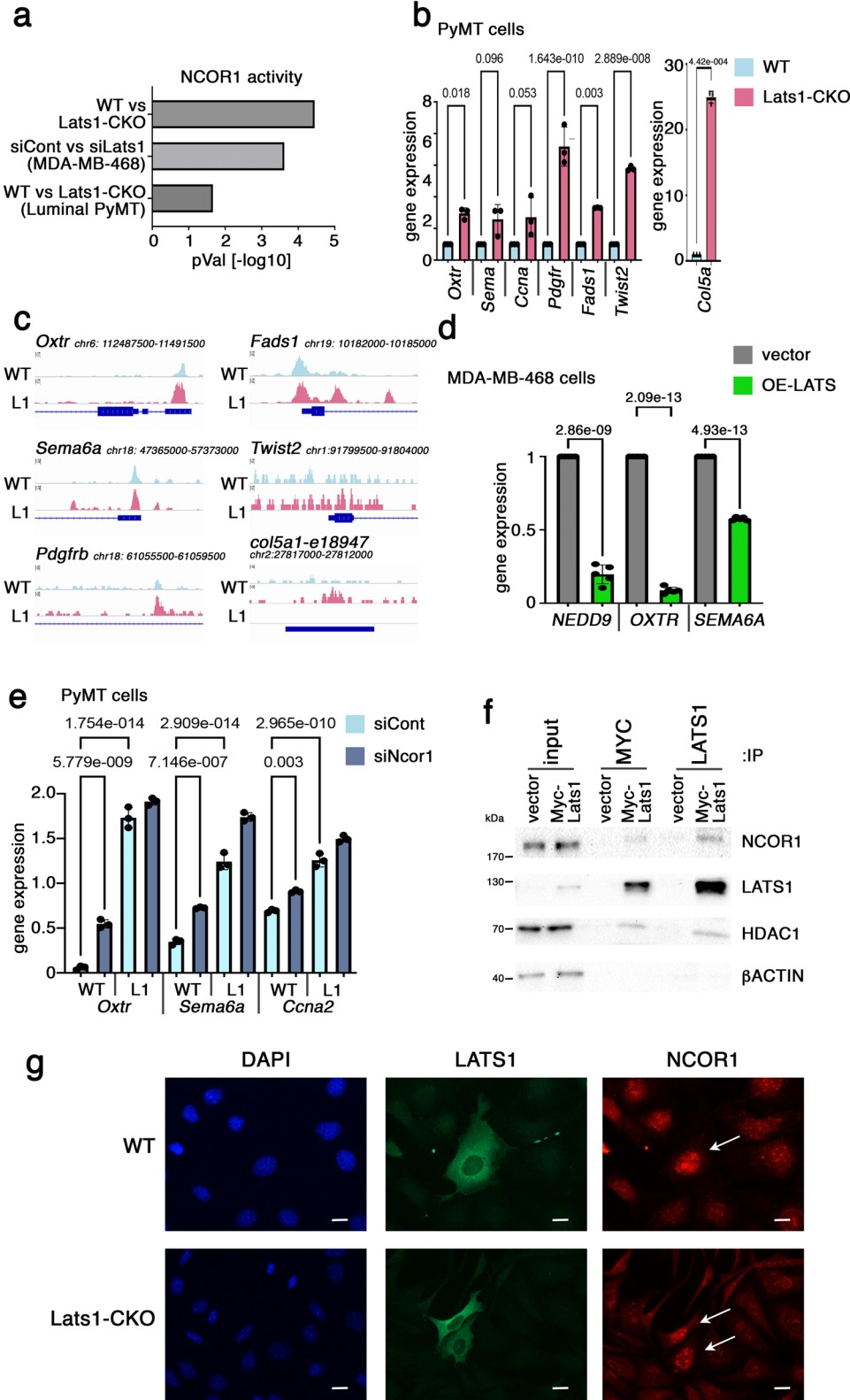

oncogenes such as receptor tyrosine kinases; thus, transformation by PyMT is strictly dependent on activation of PI3K[77,121], which is in tune with the fact that *PIK3CA* mutations are the most frequent oncogenic mutations in human luminal breast cancer.

Our findings are in line with the recently formulated notion that cells expressing a particular set of surface markers do not represent a fixed clonal entity, but rather are in a dynamic state that changes with cellular conditions[122]. Earlier thought of as a binary process, cell

identity can be viewed as a spectrum of hybrid states[123]. This has been well established in the context of epithelial-to-mesenchymal transition (EMT); cells exhibiting a hybrid partial EMT are more plastic compared to those that have undergone complete EMT or MET[124,125], making them "fittest" for metastasis[126]. Analogously, due to the adaptability afforded by cellular plasticity, hybrid luminal-to-basal-like transitioned cells[101] might represent a dangerously "fit" population for increased tumor aggressiveness.

**Fig. 5 | LATS1 interacts and cooperates with NCOR1. a** Significantly differentially expressed genes (FC > 1.5, *p*-value < 0.05) in each of the indicated comparisons were analyzed for "Upstream regulators" using IPA (QIAGEN). NCOR1 activity in the different conditions was determined by "Activation *Z*-score" and/or directionality of expression of genes repressed or activated by NCOR1, as determined by IPA. Source data are provided as a Source Data file. **b** RT-qPCR analysis of representative NCOR1-ERα repressed genes in RNA from WT or Lats1-CKO PyMT cells. Values were normalized to *Hprt*; WT values were set as 1.0. Average ± SE of three independent cell lines of each genotype. An unpaired *t*-test was used to calculate significance. Source data are provided as a Source Data file. **c** Integrative Genomics Viewer (IGV) snapshots depicting the ATAC-seq signal of representative NCOR1-ERα repressed genes in WT luminal (WT) compared to Lats1-CKO luminal (L1) cells. For each WT-L1 comparison, the *Y*-axis scale is identical. The associated RefSeq gene structure (or enhancer region) is presented below the tracks. **d** RT-qPCR analysis of representative NCOR1-repressed genes in RNA from MDA-MB-468 cells harboring vector control or inducible LATS1, following 48 h of doxycycline induction. Values were normalized to *HPRT*; vector control values were set as 1.0. Average ± SE of five

biological replicates. Unpaired *t*-test was used to calculate significance. Source data are provided as a Source Data file. **e** WT and Lats1-CKO PyMT cells were transfected with control siRNA (siCont) or siRNA against *Ncor1* (siNcor1). Three days after transfection, an additional dose of siRNA was administered. Three days later, cells were harvested for RT-qPCR analysis of the indicated NCOR1-repressed genes. Values were normalized to *Hprt*. Average ± SE of three biological replicates. One-way ANOVA was used to calculate significance. Source data are provided as a Source Data file. **f** Lats1-CKO PyMT cells, stably transduced with MYC-tagged mouse *Lats1* or vector control, were subjected to immunoprecipitation with antibodies against MYC-tag (9E10) or LATS1, followed by Western blot analysis with the indicated antibodies. 2.5% of each lysate was run as "input". β-ACTIN served as loading control for input. Representative blot of two biological repeats. **g** WT and Lats1-CKO cells, transiently transfected with GFP-tagged mouse *Lats1*, were subjected to immunofluorescent staining of NCOR1 and GFP. Arrows denote cells expressing transfected GFP-tagged mouse LATS1. Scale bar = 100 μm. Representative images of four biological repeats.

Epigenetic determinants, such as histone modifications and chromatin remodelers, play key roles in modulating lineage plasticity during tumor progression[3,127–129]. Leveraging the power of EpiTOF to analyze the global levels of a broad array of histone modifications in single cells, we observed that high H3K27ac characterizes cells in a basal-like state, whereas H3K36me2 is elevated in cells that display luminal features. Furthermore, our data suggest that the LATS1 tumor suppressor restrains luminal-to-basal-like plasticity through LATS1–NCOR1–HDAC1-mediated H3K27 deacetylation of ERα-repressed regions. While HDAC3 is considered the canonical partner of NCOR1[130], we show that LATS1 can recruit HDAC1 to serve as an effector of NCOR1-mediated H3K27 deacetylation.

Since H3K27ac distinguishes active from poised and inactive enhancers[131], curbing H3K27ac may be particularly important in restricting "basal-like" enhancer utilization to maintain luminal cell identity. Consequently, LATS1–NCOR1 loss increases H3K27ac and "opens" such enhancers. In line with our observations, an inverse correlation between the levels of H3K36me2 and H3K27ac has previously been shown to safeguard gene expression[119], and a regional imbalance of the two modifications has been associated with cancer[132]. Furthermore, KDM2B, a histone lysine demethylase that targets H3K36me2[133,134], couples H3K36me2 demethylation to H3K27 modifications[135–138] and functions as an oncogene in basal-like breast cancer[136]. Of note, other epigenetic mechanisms, such as deregulation of the SWI/SNF chromatin remodeling complex, also can promote a switch of ER-dependent luminal cells to ER-independent basal breast cancer[7,139], suggesting that integration of multiple epigenetic signals must function to prevent slippage from luminal to basal-like cell identity.

Interestingly, breast cancer cell plasticity can also facilitate the opposite transition from basal-like to a more luminal identity[140]. In fact, in an in vivo setting, MDA-MB-468 cells, which possess basal-like features, can undergo reprogramming to contribute to normal mammary gland development and generate ER+ luminal progeny[141], through alleviation of epigenetic silencing of the estrogen receptor gene *ESR1*[142]. Likewise, we found that overexpression of LATS1 in MDA-MB-468 cells appears to be sufficient to reinstate the repression of basal-like-associated genes and render luminal genes more accessible to the transcription machinery.

In line with the above observations, YAP and TAZ can repress *ESR1*, and thus specifically inhibit the growth of ER+ breast cancer cells[68]. Moreover, forced expression of TAZ in normal luminal cells induces them to adopt basal-like characteristics, and depletion of TAZ in basal-like cells leads to luminal differentiation[143]. Additionally, NCOR1 can antagonize TAZ transcriptional function in breast cancer cell lines[144]. Together, these studies suggest that in some systems TAZ may overcome NCOR1-mediated repression to impose a basal-like phenotype. It was therefore unexpected that depletion of *Taz* or *Yap* failed to

prevent the basal-like phenotype of *Lats1*-deleted luminal cells. This might be due to functional redundancy and mutual compensation of YAP and TAZ in PyMT breast tumors. Alternatively, cell identity changes upon *Yap* or *Taz* depletion may take longer than two weeks to become evident. Notwithstanding, LATS1 may conceivably prevent cell lineage plasticity in a two-pronged fashion, through both Hippo-dependent and -independent mechanisms: augmenting the pro-luminal repressive function of ERα through NCOR1-HDAC1 on one hand, and inhibiting TAZ/YAP driven luminal-to-basal-like phenotypic transition on the other hand.

The plastic nature of cancer cells is of critical importance in the clinical setting since it implies that targeting the general tumor population is insufficient to eradicate the disease[145]. Fortunately, modulating epigenetic marks to prevent plasticity offers new therapy opportunities. Boosting LATS1–NCOR1–HDAC1 activity may prevent luminal-to-basal-like transition, which may be clinically beneficial in itself, but importantly might maintain luminal-specific sensitivity to anti-hormone treatments. This is consistent with our previous observation that Lats1-CKO mammary tumors tend to develop resistance to tamoxifen[61]. We propose that patients with luminal tumors harboring low levels of *LATS1* or *NCOR1* might be sensitized to tamoxifen in combination with epigenetic treatments that "lock" the luminal state. Altogether, our findings underscore the importance of understanding the molecular mechanisms underlying cell plasticity in order to achieve effective eradication of all tumor cell subpopulations.

## Methods

Our research complies with all relevant ethical regulations. All mouse experiments were approved by the Institutional Animal Care and Use Committee (IACUC) of the Weizmann Institute (approval #06320720-2). Immunohistological assays of breast cancer patient samples in this study complied with the Weizmann Institutional Review Board (IRB) approval.

A list of materials is detailed in Table S1.

### Mice

Mice were housed in a pathogen-free facility in single-unit cages with 12-h alternate light and dark cycles and at controlled ambient temperature (21–23 °C) with humidity between 40% and 60% with free access to water and irradiated food. For PyMT tumor samples, tumors were harvested from five individual 3.5-month-old FBV/N-PyMT female mice (purchased from Jackson, strain #002374). The maximum tumor size (10% of body weight) permitted by the Animal Ethics Committee was not exceeded.

For PyMT cell injections, PyMT cells were FACS-enriched for luminal (EpCAM^high^/CD49f^high^) or basal-like (EpCAM^low^/CD49f^high^) sub-populations. $2 \times 10^6$ cells were resuspended in 100 μl sterile PBS and

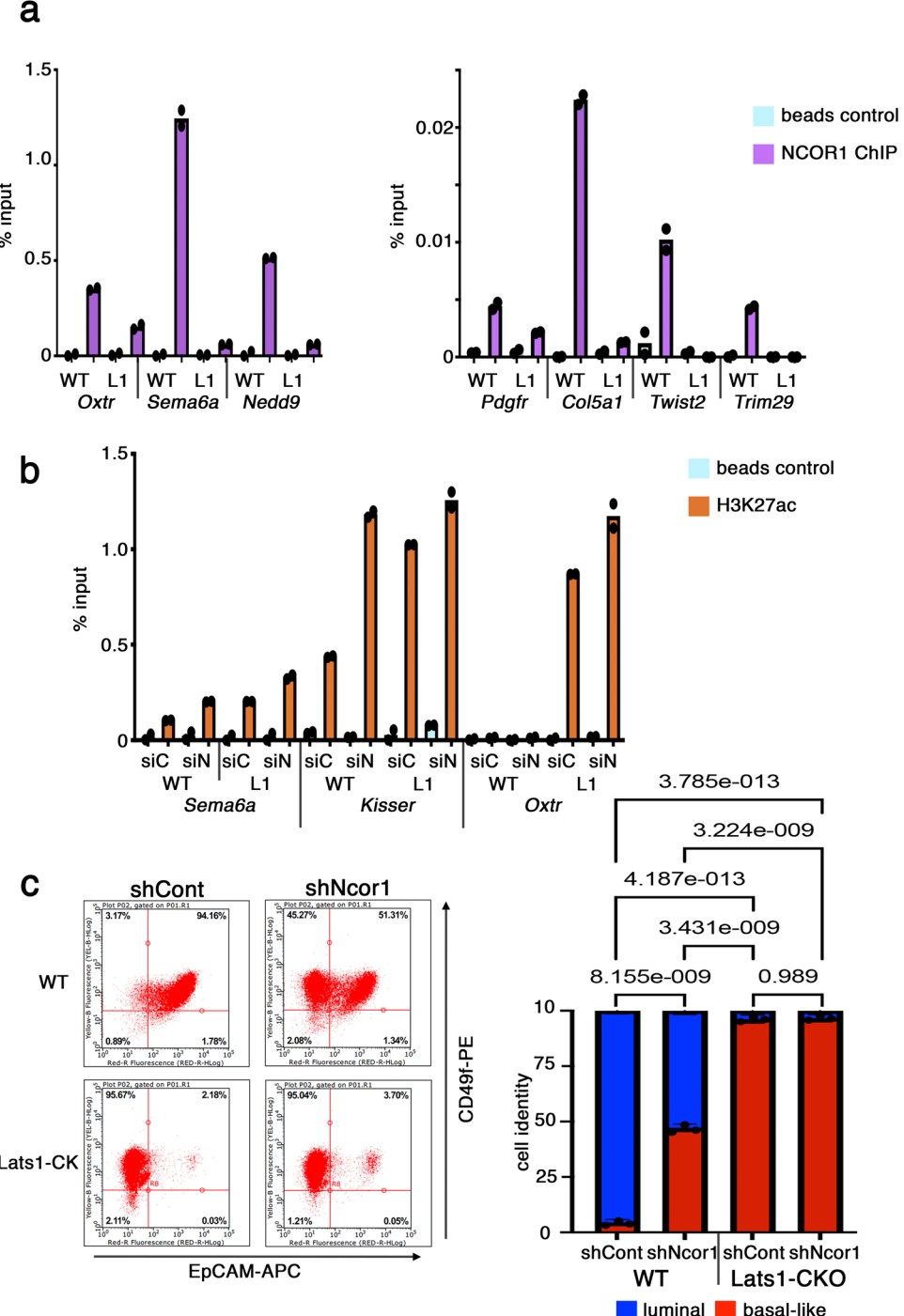

**Fig. 6 | LATS1 augments the functional recruitment of NCOR1 to ERα-NCOR1 repressed genes. a** Chromatin was immunoprecipitated from WT or Lats1-CKO PyMT cells using antibodies against endogenous NCOR1, followed by qPCR analysis of regulatory regions of the indicated LATS1–NCOR1-repressed genes. Values were normalized to input. Beads without antibodies, incubated with chromatin, served as background control. Values represent the average of two biological replicates. Source data are provided as a Source Data file. **b** WT or Lats1-CKO PyMT cells were transfected with control siRNA (siC) or siRNA against *Ncor1* (siN) for 48 h. Chromatin was immunoprecipitated using antibodies against H3K27ac, followed by

qPCR analysis of regulatory regions of the indicated LATS1–NCOR1- repressed genes. Analysis was as in (**a**). Average of two biological replicates. Source data are provided as a Source Data file. **c** WT and Lats1-CKO cells were infected with recombinant lentiviruses expressing control shRNA (shCont) or shRNA against *Ncor1* (shNcor1) and maintained under selection for at least 2 weeks. Representative FACS profiles are presented (left). A graphical representation of the relative portions of luminal (high EpCAM) and basal-like (low EpCAM) cells is shown on the right (mean ± SE of three biological replicates). One-way ANOVA was used to calculate significance. Source data are provided as a Source Data file.

injected into the mammary fat pad of 10-week-old FBV/N females (purchased from Envigo, #118). After 4 weeks, mice were sacrificed. Tumors and lungs were excised and samples were prepared for histological analysis and evaluation.

**EpiTOF**

PyMT tumor samples (0.5 g/sample) were dispersed into single-cell suspensions prior to analysis. Mechanical and enzymatic dissociation was performed using the soft tumor dissociation protocol on a

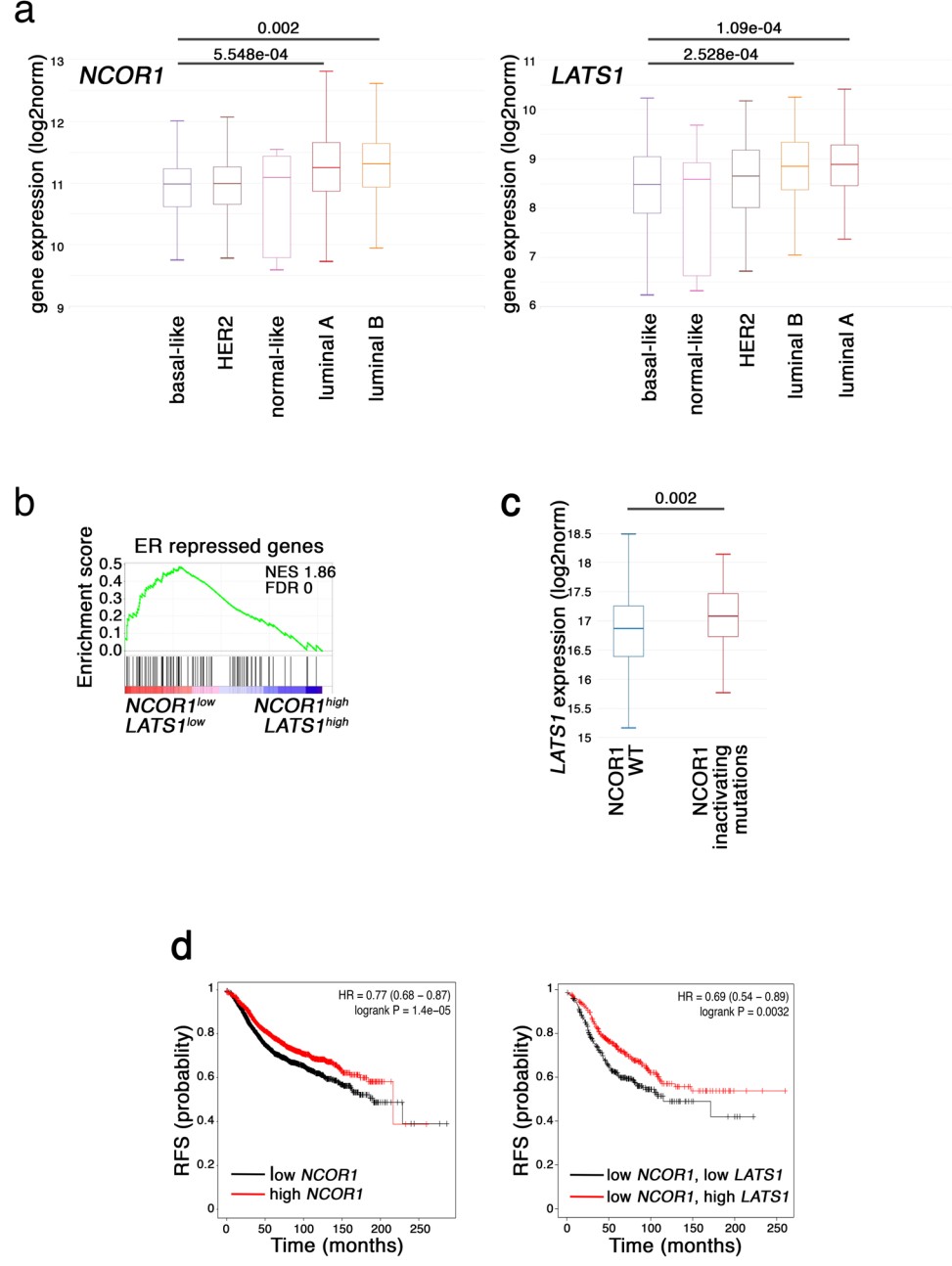

**Fig. 7 | LATS1 and NCOR1 exert similar effects on the expression of ER-repressed genes in human luminal breast cancer. a** Expression of *NCOR1* (left) or *LATS1* (right) in different breast cancer molecular subtypes. Expression (normalized log2(norm_count + 1)) was extracted from the TCGA-BRCA dataset and depicted using the XENA webtool (*n* = 522). One-way ANOVA and Tukey's post hoc test were performed. Box plots show the center line as the median, box limits as upper and lower quartiles, and whiskers as minimum and maximum values. **b** GSEA analysis of genes upregulated (ranked according to FC) in breast cancers harboring low levels of *NCOR1* and *LATS1* (NCOR1^low^LATS1^low^, bottom quartiles), compared to tumors with high levels of both genes (NCOR1^high^LATS1^high^, top quartiles), against a dataset of ER-repressed genes[30]. **c** *LATS1* expression [log2(fpkm-uq + 1)] in breast cancer tumors (TCGA-BRCA) harboring WT *NCOR1* (*n* = 919) or inactivating mutations in *NCOR1* (*n* = 37). Inactivating mutations were evaluated using PolyPhen (https://www.ensembl.org/info/genome/variation/prediction/protein_function.html). Box plots as in (**a**). Unpaired two-tailed *t*-test was performed. **d** Kaplan–Meier plots (https://kmplot.com/analysis/) of relapse-free survival (RFS) of ER+ breast tumors associated with low vs high expression of *NCOR1* (low, *n* = 1887; high, *n* = 1881) and/or *LATS1* (low, *n* = 354; high, *n* = 353).

GentleMACS Dissociator and the Multi Tissue Dissociation Kit 2 (37C_M_TDK_2, Miltenyi) according to manufacturer instructions (~40 min, starting from thawing the cryopreserved tissue to creating single-cell suspensions). After tissue dissociation, single-cell suspensions were filtered through 70 μm meshes. PyMT tumor samples (3 × 10^6 cells/sample) were cryopreserved in heat-inactivated fetal bovine serum (FBS) with 10% DMSO in liquid nitrogen. Samples were thawed rapidly into DMEM (Gibco, Invitrogen) and immediately processed.

For PyMT cell lines, single-cell suspensions were prepared by trypsin treatment (Biological Industries) of the adherent cultures for 5 min at 37 °C followed by one wash of the single-cell suspensions with complete media.

The number of viable cells in the single-cell suspensions was assessed using Trypan Blue. Isolated cells were washed with Maxpar PBS (Fluidigm #201058) and then labeled with 1.25 μM Cell-ID−Cisplatin (Fluidigm #201064) for one minute to stain for dead cells. The cisplatin was then quenched with DMEM + 10% FBS. After washing with Maxpar

Cell Staining Buffer (Fluidigm # 201068), about $3 \times 10^6$ cells per sample were incubated with the extracellular antibodies cocktail for 15 min at RT. Then, the cells were washed with Maxpar Cell Staining Buffer, fixed and permeabilized with the Maxpar Nuclear Antigen Staining Buffer Set (Fluidigm #201063) followed by barcoding with the Cell-ID 20-Plex Pd Barcoding Kit (Fluidigm #201060) according to manufacturer instructions. The cells were then washed with the Maxpar Nuclear Antigen Staining Buffer Set permeabilization buffer and the barcoded samples were combined, and then incubated with the signaling/epigenetic antibodies cocktail for 15 min at RT. Cells were then washed with Maxpar Cell Staining Buffer and fixed with fresh 4% formaldehyde (Thermo Fisher Scientific #28908) at 4 °C overnight with gentle rocking to prevent clumping. The formaldehyde solution was then supplemented with Cell-ID Intercalator-Ir (Fluidigm #201192A) at a final concentration of 125 nM and incubated for 30 min at RT to label DNA. The cells were then washed with Maxpar Cell Staining Buffer followed by Maxpar Water (Fluidigm #201069), resuspended in 1:10 EQ Four Element Calibration Beads (Fluidigm #201078) in Maxpar Water, at a concentration of about 250K cells/ml, and filtered through a 35 μm mesh. The data was acquired via a Fluidigm CyTOF Helios platform. Normalization and data cleanup to gate for the live single cells were done as in Bagwell et al.[146]. Metal conjugated antibodies were purchased from Fluidigm or conjugated in-house using the Maxpar X8 Antibody Labeling Kit (Fluidigm). The mass cytometry antibody panel was designed so that markers had a minimal signal spillover using the Maxpar Panel Designer (Fluidigm).

Analysis of the EpiTOF data was performed using an R-based pipeline described in Nowicka et al.[147]. Briefly, data were imported into R (version 4.0.2) and transformed using arcsinh with a cofactor of 5. Cell clustering was performed using FlowSom (version 1.20.0) and ConsensusClusterPlus (version 1.52.0). LATS1 was excluded from the clustering in both in vivo and in vitro experiments. In each experiment, additional marker(s) were excluded from the clustering, due to low signal. In the in vivo experiment, OCT3/4 was excluded, and in the in vitro experiment, OCT3/4 and AREG were excluded. The cells were separated into 20 clusters in the in vivo experiment and 25 clusters in the in vitro experiment. In each experiment, the clusters were manually annotated based on the relative expression of cell identity markers EpCAM, CD49f, SMA, CD24, CD44, CD45, and LATS1 and were merged to produce the final clusters. The clusters were visualized in two-dimensional space using uniform manifold approximation and projection (UMAP), implemented in the CATALYST package (version 1.12.2). The UMAP is based on the same features that were used for the clustering.

The vioplot R package (v0.3.7) https://github.com/TomKellyGenetics/vioplot was used to create the split violin plots. The data was centered for each marker, to have a zero mean.

For the contour plots, the arcsinh transformed values were standardized to have for each antigen zero mean and unit standard deviation. EpCAM and CD49f are shown on the $X$ and $Y$ axes. The contours display the relative cell frequency using the ggplot2 package https://ggplot2.tidyverse.org.

## Cell lines, transfections, and infections

All cell lines were maintained at 37 °C with 5% $CO_2$. PyMT-derived cell lines were generated from freshly minced tissue after digestion and dissociation with Gentle MACS, as described above. After dissociation, cells were filtered through 70 μm strainers, incubated with 8.58 g $NH_4Cl$ /liter Tris (pH 7.2) to lyse red blood cells, washed with DMEM, and resuspended in DMEM. The following day, adherent cells were washed vigorously to detach fibroblasts. Initially, cells were cultured in DMEM supplemented with 15% FBS, 2 mM glutamine, 1X non-essential amino acids, and 1% P/S. After the cultures had stabilized, they were acclimated to and propagated in DMEM supplemented with 10% FBS and 1% P/S.

Inducible LATS1 MDA-MB-468 cell lines were generated by transfection of pcDNA6/TR (Tet-repressor plasmid) together with pcDNA-4TO-Flag (empty vector) or pCDNA4TO-6xMyc-hsLATS1 (Tet inducible LATS1). From 72 h after transfection, cells were selected with 150 μg/ml Zeocin (Invitrogen) and 0.5 μg/ml Blasticidin (Invitrogen) in DMEM supplemented with 10% FBS and 1% P/S. For induction of LATS, cells were treated with 2 μM doxycycline for 48 h.

To generate MCF10A LATS1-KO cell line, cells were transfected (Xfect transfection reagent) with pSpCas9(BB)−2A-Puro (PX459, Addgene plasmid #48139), encoding gRNA targeting exon 5 of LATS1 (AGCAAGAAAAGTAGATACTA), and single cells were FACS sorted to 96-well dishes. Knockout clones were validated by Sanger sequencing and TIDE analysis to confirm the homozygous deletion. Unedited single-cell clones were used for WT control.

For siRNA-mediated knockdown, the indicated SMARTpools (Dharmacon, see Table S1) were used with Dharmafect #1 transfection reagent, according to the manufacturer's instructions. The final siRNA concentration was 25 nM in all cases. Plasmid transfections were performed using jetPRIME DNA transfection reagent (Polyplus Transfection) according to the manufacturer's instructions. Retroviral packaging was performed by jetPEI-mediated transfection (Polyplus Transfection) of HEK293T Pheonix cells with the appropriate plasmids, together with pMD2.G DNA encoding VSV-G envelope proteins (when infecting human cells). Virus-containing supernatants were collected 48 h following transfection, filtered, and supplemented with 8 μg/ml polybrene. Infected PyMT cells were selected with 2 μg/ml Puromycin.

## FACS procedures

Single-cell suspensions were incubated with EpCAM-APC (Miltenyi #130-102-234, 1:100) and CD49f-PE (Miltenyi #130-119-767, 1:500) for 10 min in dark at 4 °C. Unstained samples and incubation with each antibody separately served as controls. Cells were washed and resuspended in FACS buffer (0.5% BSA and 2 mM EDTA in PBS) and immediately analyzed using Guava EasyCyte (Milipore). When indicated, samples were sterilely FACS separated into tubes containing complete medium, on a FACSAria III instrument (BD Biosciences) equipped with a 407, 488, 561, and 633 nm lasers, using a 100 μm nozzle, controlled by BD FACS Diva software v8.0.1 (BD Biosciences). Further analysis was performed using FlowJo software v10.2 (Tree Star).

## Imaging flow cytometry (ImageStream)

Cells were collected with trypsin, washed with PBS, and fixed in 3.5% PFA followed by permeabilization with 0.1% Triton. Washes were done in PBS supplemented with 1% FCS and 2 mM EDTA. Cells were incubated with the indicated primary antibody for 1 h at room temperature, followed by washes and 45 min of incubation with fluorescent-conjugated secondary antibody (GaR Alexa 647, #A21244, 1:200, Thermo Fisher or GaM Alexa 595, #A11032, 1:200, Thermo Fisher) and DAPI (#D1306; LifeTech). The cells were imaged by ImageStreamX Mark II (Amnis, part of EMD Millipore) using bright-field 488, 561, and 642 nm lasers. At least 30,000 cells were collected from each sample and data was analyzed using image analysis software (IDEAS 6.2; Amnis Corporation). Images were compensated for fluorescent dye overlap by using single-stain controls. Gating was done for single cells, using the area and aspect ratio features, and for focused cells using the gradient RMS feature, as previously described[148]. Data were analyzed with the IDEAS 6.1 software (Amnis, part of EMD Millipore). Only cells with an intact nucleus (according to DAPI staining) were analyzed. Nuclear localization was determined by the similarity feature on the nuclear mask of the DAPI staining and the relevant antibody signal (the log-transformed Pearson's correlation coefficient in the two input images). Similarity >1.5 was considered significant. Positively stained cells were gated on the basis of comparison with a nonstained sample.

## Isolation of total RNA, reverse transcription and RT-qPCR

RNA was isolated using the NucleoSpin kit (Macherey Nagel), RNeasy Mini kit (Qiagene), or RNeasy Microkit (Qiagen). 1–2 μg of each RNA

sample was reverse-transcribed using Moloney murine leukemia virus reverse transcriptase (Promega) and random hexamer primers (Applied Biosystems). Real-time qPCR was performed using SYBR Green PCR supermix (Invitrogen) with a StepOne real-time PCR instrument (Applied Biosystems). For each gene, values for the standard curve were measured and the relative quantity was normalized to *HPRT* or *GAPDH* mRNA.

## Co-immunoprecipitation (co-IP) analysis

Cell monolayers were gently washed twice with ice-cold PBS and lysed on ice for 2 h with NP-40 lysis buffer (50 mM Tris–HCl pH 8.0, 150 mM NaCl, 1.0% NP-40) supplemented with protease inhibitor mix (Sigma) and phosphatase inhibitor cocktail I + II (Sigma). Protein A Dynabeads (catalog no. 10002D), pre-incubated with appropriate antibodies (1:100) 24 h prior to IP, were added to the cleared lysates and incubated by rotating for 4 h at 4 °C. Immunoprecipitates were washed twice with NP-40 lysis buffer, collected by DynaMag-2 (catalog no. 123.21D), released from the beads by boiling and resolved by SDS–PAGE. Western blots were imaged using ImageLab (v4.1).

## Western blots

Cell pellets were resuspended in a protein sample buffer and boiled. Samples were resolved by SDS–PAGE. Panels probed for proteins of similar molecular weight, such as comparisons of different histone H3 modifications, or visualization of Myc-tagged LATS1 with anti-Myc tag antibody (Abcam #ab32, 1:1000) vs. anti-LATS1 antibody (CST, #3477, 1:1000), were run in separate lanes of the same gel, using identical amounts of lysate. Equal loading was confirmed by comparison to GAPDH (CST, #2118, 1:1000) levels in each lane. Imaging was accrued using a ChemiDoc MP imaging system (BioRad) with the Image Lab 4.1 program (BioRad).

## Indirect ChIP

Cell monolayers were gently washed with ice-cold PBS, fixed on the culture dish in 5 mM DTBP in ice-cold PBS for 30 min, and then further fixed with 1% formaldehyde (Thermo Scientific, #28908) and incubated for an additional 20 min at RT. Fixation was stopped with 0.125 M Glycine followed by incubation for 5 min at RT. Cells were washed, harvested, and resuspended in cell lysis buffer (5 mM PIPES, pH 8, 85 mM KCl, and 0.5% NP-40). Diluted samples were sonicated to obtain DNA fragments of 150–600 bp. After centrifugation (10 min at 4 C), samples pre-cleared with beads only were split for incubation with antibody (25%) and retention for input (4%). Chromatin was rotated with antibodies overnight at 4 °C. The following day immunoprecipitates were rotated for 2 h at 4 °C with Dynabeads magnetic beads (Invitrogen #10003D). Subsequently, beads were washed and then washed and resuspended in TE. Crosslinking was reversed with RNase and Proteinase K. The following day, DNA fragments were isolated using a PCR purification kit (QIAGEN).

## Immunofluorescence

Cells were plated and grown on 12 mm coverslips. 24 h later, cells were gently washed twice with cold PBS and fixed with PFA 3% in PBS for 20 min at RT. After an additional wash with PBS, samples were permeabilized (Triton X-100 0.1% in PBS, 5 min at RT) and then blocked with 5% FCS in PBS. Samples were then incubated overnight with primary antibody, washed, and then incubated with secondary antibody and DAPI (5 mg/ml final) for 60 min in the dark.

## In situ proximity ligation assay

Cells were fixed with 4% PFA for 15–20 min and permeabilized with 0.1% Triton for 5 min. PLA was performed using the DuoLink In Situ PLA Detection Kit (DUO92101, Sigma). Imaging was done using an LSM 800 (Zeiss) confocal microscope with ×40 or ×60 objective oil immersion.

The following antibodies were used for PLA: GFP (Abcam #ab1218, 1:200), NCOR1 (Cell Signaling #5948, 1:100).

## ATAC-seq

Sample preparation was conducted as previously described by Buenrostro et al.[149], with modifications described by Lara-Astiaso and colleges[150]. Briefly, 50,000 cells from WT luminal, WT basal-like, Lats1-CKO luminal, or Lats1-CKO basal-like cultures were used (two replicates each). Nuclei were incubated with 2 μl of Nextera Tn5 enzyme (TDE1, Illumina) for 1 h at 37 °C. Enzyme inactivation was done by the addition of 5 μl Clean-up buffer (900 mM NaCl, 30 mM EDTA), 2 μl of 5% SDS, and 2 μl of Proteinase K (NEB) and incubation for 30 min at 40 °C. Tagmented DNA was isolated using 2× SPRI beads cleanup.

For library amplification, two sequential 9-cycle and 5-cycle PCR were performed in order to enrich small tagmented DNA fragments. Libraries were prepared using KAPA HiFi HotStart ready mix. After the first PCR, the libraries were selected for small fragments using SPRI cleanup (0.65×). Then a second PCR was performed with the same conditions in order to obtain the final library. DNA concentration was measured with a Qubit fluorometer (Life Technologies) and library sizes were determined using TapeStation (Agilent Technologies). Libraries were sequenced on the NovaSeq6000 sequencing platform using SP, 100 cycles kit (paired-end sequencing), with an average of 120 million reads obtained for each sample.

Adapters were trimmed using the cutadapt tool. Following adapter removal, reads shorter than 30 nucleotides were discarded (cutadapt option –m 30). Reads were aligned uniquely to the mouse genome (mm10) using bowtie (version 1.0.0)[151]. Reads that mapped to mitochondrial DNA were excluded. Duplicate reads were excluded using Picard tools. Fragments longer than 120 were excluded from the analysis. Open regions (peaks) were detected using MACS2 (version 2.0.10.20131216)[152]. Peaks overlapping with the Encode mm10 blacklist were excluded using bedtools. Peaks overlapping with the Encode mm10 blacklist were excluded using bedtools. Differences between WT luminal and WT basal samples were inferred using DiffBind (http://bioconductor.org/packages/release/bioc/html/DiffBind.html) tool: for each region, peak concentration was defined as log2 normalized read count and the log2 fold change was calculated. Differential peaks were defined using raw $p$-value < 0.05 as the significance cutoff. Replicate bam files were combined using samtools merge function[153] (beside L1-CKO luminal sample, in which one replicate was discarded due to quality control) (Table S2) and bigwig files were generated using DeepTools2[154] (--exactScaling --binSize 1 --normalizeUsing CPM --extendReads). Reads coverage around TSS was visualized using ngs.plot[155]. DeepTools2[154] was used to generate heatmaps and profiles. GREAT[156] and HOMER[157] (annotatePeaks.pl) were used to associate peaks with genes (±5 kb from TSS). Promoter–enhancer interactions were downloaded from ENC+ EPD Enhc-Gene track (UCSC table browser[158]). Enrichment of known motifs was calculated using the "overrepresented TFBS" tool of the Genomatix Software Suite[159]. Additional ERE motif was extracted from HOMER Motif Database[157].

## MARS-seq

RNA-seq libraries were prepared at the Crown Genomics Institute of the Nancy and Stephen Grand Israel National Center for Personalized Medicine, Weizmann Institute of Science. A bulk adaptation of the MARS-Seq protocol[160] was used to generate RNA-seq libraries for expression profiling. Briefly, 30 ng of input RNA from each sample was barcoded during reverse transcription and pooled. Following Agencourt AMPure XP beads cleanup (Beckman Coulter), the pooled samples underwent second-strand synthesis and were linearly amplified by T7 in vitro transcription. The resulting RNA was fragmented and converted into a sequencing-ready library by tagging the samples with Illumina sequences during ligation, RT, and PCR. Sequencing was done

with a Nextseq 75 cycles high output kit (Illumina) and analyzed as follows[161]. Reads were trimmed using cutadapt (http://code.google.com/p/cutadapt/) and mapped to the mm10 genome using STAR[162] v2.4.2a (default parameters). The pipeline quantifies the genes annotated in Gencode (that have been expanded with 1000 bases toward the 5' edge and 100 bases toward the 3' bases). Counting was done using htseq-count[163] (union mode). Further analysis was done only for genes having a minimum of 5 reads in at least one sample. Normalization of the counts and differential expression analysis was performed using DESeq2[164] with the parameters: betaPrior = True, cooksCutoff = FALSE, independentFiltering = FALSE. Differentially expressed genes were defined using FC > 1.5 and raw $p$-value < 0.05 as a significance cutoff.

### Functional analysis of gene expression data

For GSEA analysis[165], genes were ranked according to fold change between the two described conditions, with only significant differences considered ($p$-value < 0.05). Comparison to different genesets was done using GSEA preranked tool. A similar ranking was used to analyze gene expression patterns by the Ingenuity Pathway analysis software (QIAGEN Inc., https://www.qiagenbioinformatics.com/products/ingenuitypathway-analysis).

### Ingenuity pathway analysis

Expression patterns (log2) between the indicated conditions were analyzed using the Ingenuity Upstream Regulator analysis of IPA[111] (QIAGEN Inc., https://www.qiagenbioinformatics.com/products/ingenuitypathway-analysis). Expression difference significance cutoff (pre-filtering) was set to either p-value<0.05 (Luminal WT-PyMT) or adjusted $p$-value < 0.05 (MDA-MB-468 and tumors[61]).

### Statistics and reproducibility

Three independent biological replicates were performed unless otherwise stated. Statistical analysis was performed using the GraphPad Prism 9.1.0 software unless otherwise stated.

### Reporting summary

Further information on research design is available in the Nature Portfolio Reporting Summary linked to this article.

## Data availability

The RNA-Seq and ATAC-Seq data generated in this study were aligned to mouse genome assembly (mm10) and have been deposited in NCBI's Gene Expression Omnibus database and are accessible through GEO Series accession number GSE195716. The CyTOF data was uploaded to the FlowRepository, FR-FCM-Z5L5 and FR-FCM-Z5L6. All unique materials used in this study are readily available from the authors or from standard commercial sources. Source data are provided with this paper.

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

## Acknowledgements

We thank James Martin (Baylor College of Medicine, Houston, TX, USA) for the generous sharing of conditional knockout mice. We thank Dr. Ron Rotkopf (Bioinformatics Life Science Core Facility, Weizmann Institute, Israel) for bioinformatic help and Ziv Porath (Flow Cytometry Unit, Department of Biological Services, Weizmann Institute, Israel) for assistance with ImageStream analyses. Dr. Werner Schroth (University of Tuebingen, Stuttgart, Germany) who provided us with the breast cancer FFPE blocks and patient data from his study collection, and Ms. Kerstin Willecke (University of Tuebingen, Stuttgart, Germany) for performing IHC staining on the human patient samples. This work was supported in part by Dr. Miriam and Sheldon G. Adelson Medical Research Foundation (Grant G-201906-00324 to M.O.), the Robert Bosch Stiftung GmbH, and the Berthold Leibinger Stiftung GmbH (Grant 123493 to M.O.), the Rising Tide Foundation (Grant 136467 to M.O.), the United States–Israel Binational Science Foundation (BSF), Jerusalem, Israel (Grant 2019045 to M.O.), Anat and Amnon Shashua (Grant 721972 to M.O.), and the Moross Integrated Cancer Center. M.O. is the incumbent of the Andre Lwoff chair in molecular biology. G.F. is incumbent of the David and Stacey Cynamon Research fellow Chair in Genetics and Personalized Medicine. E.S is supported by the Emerson Collective and the Israel Cancer Research Fund, and is an incumbent of the Lisa and Jeffrey Aronin Family Career Development chair.

## Author contributions

Y.A. and M.O. designed research; Y.A. and N.F. performed research; G.M., N.B.N., M.D., O.H., R.Z., B.C., V.D., T.M.S., S.M., and N.H. helped with the experiments; N.F. and G.F. helped with the analyses; R.J., W.E.A., Y.Y., E.S., and M.O. supervised research; Y.A., N.F., and M.O. wrote the paper. All authors discussed the results and commented on the manuscript.

## Competing interests

The authors declare no competing interests.
