## [Peer Review File · Nature Communications]

Breast cancer plasticity is restricted by a LATS1-NCOR1 repressive axisREVIEWER COMMENTS

Reviewer #1 (Remarks to the Author):

Aylon et al have presented an innovative strategy to identify a non-conventional role of LATS1 in maintaining luminal cell plasticity integrating multiple approaches. The authors have extensively shown the contribution of NCOR1 to LATS1 utilizing biochemical assays and proximity ligation assay. The authors have revealed the chromatin state in luminal and basal cells in the context of LATS1 cKO, utilizing ATAC-Seq to address cellular plasticity. The authors have beautifully presented the accessibility state in basal and luminal cells utilizing the LATS1cKO mice. They concluded with clinical significance of NCOR1-LATS1 axis in luminal breast cancer patients.

Comments:

The authors have generated ATAC-Seq data from the basal and luminal cells utilizing WT and LATS1 CKO mice. The authors should have done MOTIF analysis utilizing the regions that either gained or lost accessibility in the basal and luminal cells. The authors should try looking at the regions and ran GSEA analysis that identified ERalpha activation or repressed genes. However the results do not reveal the actual fold change in the accessibility of these genes that enriched in the GSEA analysis. The authors have also performed RNA-Seq utilizing the cells from WT and CKO. The authors should integrate the ATAC-Seq and RNA-Seq to find if the change in accessibility contributed to gene expression changes? This will also help the authors in identifying novel targets, if any, rather than the few genes that were validated in figures 5 and 6 (OXTR, SEMA6A, NEDD9, KRT18, KRT14).

The few genes that the authors picked and validated in the context of LATS and NCOR1 (figs 5 and 6) were validated only at the expression level. Since the authors have already done the ATAC-Seq, it would have been nice to have the accessibility changes at the genes in luminal and basal cells from WT and LATS1 CKO cells.

The authors have indicated overall changes in H3K36me2 and H3K27ac in figure 1B. They also validated it by western blotting. Changes in H3K27ac was analyzed at few target genes, OXTR, KLH19, KISSER, and SEMA6A. However the H3K36me2 was not analyzed. It would be nice to evaluate the H3K36me2 status as well.

In Figure 5A, the authors mention NCOR1 activation. However they do not mention how it was assessed. The assumption would be the gene list that the authors cite from qiagen/IPA might have been utilized. But it should be clearly indicated in the text.

The authors have done a very careful and controlled experiment as shown in Figure 3, wherein they show how basal cells when passaged or injected into mammary fat pad results in enriching luminal cells. It would have been interesting to analyze the expression of the candidate genes as well as any novel genes they would identify by integrating the ATAC-Seq and RNA-Seq data in the multiple passages across multiple panels, in Figure 3A (panel 2, 4, 5, 7). Does their analysis also suggest that basal cells in WT condition with LATS1 expression will give rise to luminal cells? and can it be therapeutically targeted integrating the NCOR1 complex or p300 inhibitors which promote the H3K27ac?

Minor comments:

The authors should calculate and indicate the percentage of population in their FACS data like how they have indicated in Figure 6C.

Changes in histone modification as shown in westerns does not reflect the actual changes on some promoters. Global ChIP-seq or at the least changes in H3K36me2 at the target genes would have been informative as the authors start the manuscript in Figure 1, highlighting the changes in histone modifications which was very selective to only H3K36me2 and H3K27ac.

The authors mention utilizing their RNA-Seq data and mention that the NCOR1 activity has been compromised and it is not clear if it is based on pathway analysis of the genes, or the expression of

NCOR1 itself. If it was pathway analysis, do they see the epigenetic modifier that contributes to H3K36me2 in the analysis. Motif analysis of their ATAC-seq would have given novel insights.

Reviewer #2 (Remarks to the Author):

In the present manuscript "Breast cancer plasticity is restricted by a LATS1-NCOR1 repressive axis" Aylon et al report that LATS1, helps maintain "a luminal identity" of PyMT mammary carcinoma cells by keeping "basal-specific" genes in a closed chromatin state through interaction with the nuclear corepressor NCOR1 and recruitment of HDAC1, driving histone H3K27 deacetylation.

The authors previously demonstrated using the MMTV-polyomavirus middle T oncogene (PyMT) driven mammary carcinoma model that deletion of LATS1 results in increased expression of "basal-like features" in the tumors with a reduced response to tamoxifen (Furth et al). Here, they provide molecular underpinnings for this phenomenon in the PyMT-driven mammary carcinoma cells. Underlying the increased expression of basal features are epigenetic changes that involve opening of chromatin in regions encoding "basal genes". The authors provide evidence that this phenomenon is YAP/TAZ independent. By showing that a Myc-tagged mouse LATS1 co-immunoprecipitates with endogenous NCOR1 in PyMT-carcinoma cells and that HDAC1 and NCOR1 only interact in the presence of LATS1 pointing to a ternary repressive complex that silences "basal genes".

Major concerns:

The data presented are clean and sound but the imprecise use of breast cancer-related terminology and the inadequacy of the models severely limit the biological interest of this work. The main claim about "breast cancer plasticity" is not supported by the data

The authors assertion that decreased expression of LATS1 increases expression of basal genes and drives breast cancer cells (implying the human disease) into a basal program and tumor progression when the experimental evidence is merely based on the PyMT model. Ectopic expression of this viral oncogene results in very fast growing tumors that bear little resemblance to the human disease.

The polyomavirus middle T oncogene (PyMT) induced mammary carcinomas express the estrogen receptor only in the early stages of tumorigenesis. Tumor growth has not been rigorously shown to be hormone-dependent; i.e. response to ovariectomy. Hence, the model reflects, if at all, human luminal B tumors only very initially. It is not clear at what age and stage of tumor development the experiments were performed.

The terms "luminal" and "basal" designate the two major cell lineages in the normal mouse mammary epithelium and the normal human breast epithelium. The nomenclature for breast cancer subtypes is "luminal A or B" and "basal-like". As the authors point out the latter term is misleading as basal-like breast carcinomas were shown to arise from luminal progenitor cells. Hence all breast carcinomas arise from luminal cells.

This imprecision in terminology confounds this manuscript. It is unclear what basal and luminal genes are: do the authors refer to the normal cell types in the breast, the mouse mammary epithelium, or are they related to molecular tumor subtypes? Moreover, these terms are intermingled with ER signatures.

Minor comments :

FACS analysis: distinction of basal and luminal tumor cells is merely based on EpCAM expression levels. This is misleading when luminal tumors are ER+ and often progesterone receptor + while basal-like tumors lack hormone receptor expression. There is no reference to EpCAM expression levels distinguishing between tumor subtypes. CD49f not expressed in mature luminal and hormone receptor positive mammary epithelial cells but here it is high in both populations.

Figure 4E: the finding that LATS1 over expression increases levels of EpCAM detected on the cell surface is not sufficient to conclude on "luminal differentiation".

Why chose MDA-MB-468 to test for the repressive effects of LATS1 overexpression?

Line 176: "both mouse and human cells" should read "both mouse mammary PyMT carcinoma cells and a human breast cancer cell line"

The signature for "ER repressed" genes is based on a publication performed with microarrays in MCF7 cells in 2003 upon short time treatment of these cells in vitro.

Figure 7B, labeling issue: x-axis.

Reviewer #3 (Remarks to the Author):

The manuscript by Aylon et al. studied the essential role of LATS1 in maintaining the luminal identity of breast tumor cells. The authors found that LATS1 interacted with NCOR1 and HDAC1 to regulate chromatin accessibility, which in turn, represses the expression of genes associated with the basal cell state. While this is an interesting paper, some conclusions were weakly supported by the data.

Major issues:

1. The data, in many cases, lack adequate biological replicates and lack sufficient statistical analysis. For example, the Western blotting, flow cytometry, and ATAC-seq only showed one sample for each group. More biologic replicates would strengthen the paper.
2. Throughout the study, the authors define cell state by only two genes, CD49f and EpCAM. More genetic markers, a gene "signature", are needed to provide more evidence of the basal or luminal cell state.
3. In Fig 5, the similar trends of gene expression in LAST1-KO or NCOR1-KO cells may be correlation only. The authors should provide more evidence for their conclusion that LATS1 exerts its function through NCOR1. For example, checking gene expression in LATS1 overexpressed cells with NCOR1 deletion.
4. The mechanistic connection between LATS1 and NCOR1 is unclear. The key data regarding the interaction of LATS1 and NCOR1 is in an overexpression context and in vitro. Also, some controls are missing from the experiments.
5. In Fig 5G, it appears that LATS1 and NCOR1 are not co-localized which argues against the two proteins forming a complex to regulate chromatin accessibility. This is also an overexpression system. Using a system with endogenous levels of proteins is more impactful. Moreover, quantification is required.

Minor issues:

1. Fig 1A, it would be better to show a bar graph indicating the percentages of different cell populations.
2. Fig 1C is confusing.
3. Fig 1E, the expression of CD49f seems to be similar in all three cell types. Please clarify.
4. Fig 2A and 2B, please clarify the method for normalizing cell numbers for the sequencing experiment.
5. Fig 2C bottom, the read counts of Basal genes were lower in WT basal cells as compared to WT luminal cells. Please clarify this
6. Fig 2E bottom. It might be better to represent the data as a heat map with representative genes.
7. Fig 3A, please show the percentages of each population. Is it that possible the luminal cells grow faster than basal cells in culture conditions?
8. Fig S4C, the authors should consider showing the levels of phosphorylated YAP/TAZ.
9. Please clarify the cell types used in Fig 6A and 6B?

We would like to thank the reviewers for their insightful comments. We believe that the additional experiments and analyses that we carried out in response to those comments have improved the paper significantly.

Reviewer #1 (Remarks to the Author):

Aylon et al have presented an innovative strategy to identify a non-conventional role of LATS1 in maintaining luminal cell plasticity integrating multiple approaches. The authors have extensively shown the contribution of NCOR1 to LATS1 utilizing biochemical assays and proximity ligation assay. The authors have revealed the chromatin state in luminal and basal cells in the context of LATS1 cKO, utilizing ATAC-Seq to address cellular plasticity. The authors have beautifully presented the accessibility state in basal and luminal cells utilizing the LATS1cKO mice. They concluded with clinical significance of NCOR1-LATS1 axis in luminal breast cancer patients.

We thank this reviewer for their positive comments on our manuscript.

Comments:

The authors have generated ATAC-Seq data from the basal and luminal cells utilizing WT and LATS1 CKO mice. The authors should have done MOTIF analysis utilizing the regions that either gained or lost accessibility in the basal and luminal cells.

*As requested, we performed an unbiased search for motif families enriched (z-score >2) in peaks associated with “luminal” and “basal-like” regions (from Fig. 2B). The inclusive list of enriched Genomatix transcription families is now included as a new **Supplementary Table S3**. To directly interrogate estrogen receptor binding motifs, most relevant to our study, we compared the significance of individual variations of estrogen response elements (EREs) available in Genomatix or HOMER databases. As observed in the **new Supplementary Table S4**, ChIP-derived EREs (V\$ER.03, V\$ER.04, V\$ESR2.01 and ERE (NR)-HOMER) all showed a significant enrichment in basal-like-specific peaks. This is in line with our model, in which ER-dependent repression of chromatin accessibility in luminal cells is important to maintain luminal cell identity. As expected, EREs were also enriched in luminal-specific peaks, consistent with the well documented transactivation function of ER in such cells.*

*Interestingly, upon scrutiny of the pattern of EREs enriched in “basal-like region” peaks, we noticed that they were predominantly located in enhancers, which was not the case for peaks in “luminal regions”. We therefore have now expanded some of our analyses to include enhancer regions. For instance, we now show that enhancer e18947, associated with the Col5a1 gene, which harbors an ERE motif, is more accessible and hence less efficiently repressed in basal-like and L1-CKO cells. Col5a1 and its associated enhancer are now included in several of our analyses (**New Fig. 5B, C, S5C, 6A and S6B**)*

The authors should try looking at the regions and ran GSEA analysis that identified ERalpha activation or repressed genes. However, the results do not reveal the actual fold change in the accessibility of these genes that enriched in the GSEA analysis.

To evaluate differential chromatin accessibility in WT and Lats1-CKO cells in the regulatory regions of genes associated with ERa activation or repression, we took two approaches. First, we associated the 21 genes contributing to the leading edge of the **ERa activation** signature enrichment in our GSEA with ATAC peaks located within 5kb of their TSS. This resulted in 40 ATAC-seq peaks for which the coverage was calculated (log2 normalized read counts). As predicted, this group of regions was significantly more “open” in WT luminal, compared to WT basal-like cells. The (non-paired) fold difference in peak concentration between luminal and basal-like cells was 1.6. The data is now included in the manuscript as *new Fig. S2F*. Second, we took an alternative approach to examine the chromatin accessibility of regulatory regions of **ER repressed** genes, based on the above observation that these regions may reside particularly within enhancers. To that end, we looked for enhancers linked to the 22 genes contributing to the leading edge of the ER repressed signature enrichment in our GSEA. This resulted in 72 enhancers linked to 14 genes of this signature. We found that these regions were more accessible in WT than in Lats1-CKO cells (*new Fig. S2H*), in line with the differential gene expression pattern shown for these genes in figure 2F. Interestingly, a similar analysis of enhancers linked to genes contributing to ERa activation signature enrichment did not display differential accessibility.

The authors have also performed RNA-Seq utilizing the cells from WT and CKO. The authors should integrate the ATAC-Seq and RNA-Seq to find if the change in accessibility contributed to gene expression changes? This will also help the authors in identifying novel targets, if any, rather than the few genes that were validated in figures 5 and 6 (OXTR, SEMA6A, NEDD9, KRT18, KRT14).

To integrate our RNA-seq and ATAC-seq data, we performed two complementary analyses. First, we plotted the ATAC-seq peaks that were within 5kb of known TSSs (a total of 297 peaks) against the RNA-seq data ($p\text{Val} < 0.05$) of the same genes, demonstrating a linear positive correlation between gene expression and chromatin accessibility (Pearson correlation coefficient of 0.46) (*now Fig. S2E*).

Second, we visualized the RNA-seq data (Lats1-CKO luminal vs. WT luminal comparison) in a volcano-plot. Significantly ($p\text{-val} < 0.05$) differential ATAC-seq peaks (within 5kb of the TSS of the same gene) were colored according to whether they were up in WT luminal (blue) or up in WT basal-like (BL) (red) (*new Fig. S2J*). This highlighted additional genes repressed by LATS1 and associated with regulatory regions that were more “open” in basal-like cells. We have now included these genes in further analyses (see *modified Figs. 5B, 5C, S5C, 6A and S6B*).

We further examined the genes differentially expressed in luminal compared to basal-like cell states, searching for accessibility changes within enhancers linked to these genes. Interestingly, we identified differentially open enhancers only in those linked to “basal” genes (*new Fig. 2D, left and right*). As expected, Lats1-CKO basal-like cells displayed the most open chromatin of these “basal” enhancers, further strengthening the notion that LATS1-NCOR1 may prevent basal-like gene expression by maintaining the chromatin, both adjacent to TSSs

and in distal enhancers, in a closed state. We include a representative IGV track for one such enhancer (Col5a-enhancer e18947) in *new Fig. 5C*.

The few genes that the authors picked and validated in the context of LATS and NCOR1 (figs 5 and 6) were validated only at the expression level. Since the authors have already done the ATAC-Seq, it would have been nice to have the accessibility changes at the genes in luminal and basal cells from WT and LATS1 CKO cells.

We now include IGV tracks of regulatory regions of genes repressed by LATS1/NCOR1. As expected, chromatin of these regions is more accessible in Lats1-CKO, compared to WT, cells (new Fig. 5C).

The authors have indicated overall changes in H3K36me2 and H3K27ac in figure 1B. They also validated it by western blotting. Changes in H3K27ac was analyzed at few target genes, OXTR, KLH19, KISSER, and SEMA6A. However, the H3K36me2 was not analyzed. It would be nice to evaluate the H3K36me2 status as well.

We performed ChIP of H3K36me2 and checked by qPCR the abundance of this modification on regulatory regions of LATS1/NCOR1-repressed genes. We observed a negative correlation between H3K27ac and H3K36me2 abundance on these genes (as noted in the revised text, lines 353-357). H3K27ac was suppressed by LATS1, whereas H3K36me2 levels were augmented in WT, compared to Lats1-CKO, cells (new Fig. S6E), supporting the notion of a functional connection between these two histone marks, as also reported previously in the literature¹⁻³.

In Figure 5A, the authors mention NCOR1 activation. However they do not mention how it was assessed. The assumption would be the gene list that the authors cite from qiagen/IPA might have been utilized. But it should be clearly indicated in the text.

We apologize for the lack of clarity of the text. The reviewer is correct. As explained in the figure legend, “Significantly differentially expressed genes (FC>1.5, p-value<0.05) in each of the indicated comparisons were analyzed for “Upstream regulators” using IPA (QIAGEN). NCOR1 activity in the different conditions was determined by “Activation Z-score” and/or directionality of expression of genes repressed or activated by NCOR1, as determined by IPA.” We now clarify this point also in the main text (lines 288-289).

The authors have done a very careful and controlled experiment as shown in Figure 3, wherein they show how basal cells when passaged or injected into mammary fat pad results in enriching luminal cells. It would have been interesting to analyze the expression of the candidate genes as well as any novel genes they would identify by integrating the ATAC-Seq and RNA-Seq data in the multiple passages across multiple panels, in Figure 3A (panel 2, 4, 5, 7).

Thank you for the interesting suggestion. We quantified the expression of one “previous” and one “new” LATS1/NCOR1-repressed “basal” gene from RNA extracted at time points when

cells were split throughout the experiment. As now shown in *new Fig. 3B*, target expression decreased as cells transitioned from a basal-like to luminal phenotype, in line with the prediction from our model.

Does their analysis also suggest that basal cells in WT condition with LATS1 expression will give rise to luminal cells?

Yes. In fact, that is the implication of the previous Fig. 4E (now Fig. 4F). In the original version of the manuscript, we also presented a GSEA demonstrating that, upon overexpression of LATS1 in basal-like MDA-MB-468 cells, a luminal, ER repressed gene signature is partially instated (previous Fig. 2E, now Fig. 2F, right).

*To further examine the ability of LATS1 to drive a luminal phenotype, we re-expressed LATS1 in Lats1-CKO cells and checked LATS1-dependent expression of the canonical luminal marker KRT8 (*new Fig. S4B*). Expression of LATS1 significantly augmented the intensity and number of cells expressing KRT8, further strengthening the notion that LATS1 promotes a luminal cell identity.*

and can it be therapeutically targeted integrating the NCOR1 complex or p300 inhibitors which promote the H3K27ac?

*We thank the reviewer for this interesting and clinically relevant suggestion. To address this possibility, we treated WT and Lats1-CKO, luminal and basal-like cells with the p300 inhibitor A-485. This strongly decreased H3K27ac, as expected (*new Fig. S6C*). Importantly, A-485 treatment reversed the effect of Lats1 depletion on gene transcription of NCOR1-repressed targets (*new Fig. S6D*). Although beyond the scope of the current work, we remain hopeful of the potential to therapeutically target the LATS1-NCOR1 axis to repress aberrant gene expression and “fix” breast cancer cells in a luminal state, curbing their escape from anti-hormone therapy.*

Minor comments:

The authors should calculate and indicate the percentage of population in their FACS data like how they have indicated in Figure 6C.

We apologize for the oversight. The percentage of population in the FACS data has been added.

Changes in histone modification as shown in westerns does not reflect the actual changes on some promoters. Global ChIP-seq or at the least changes in H3K36me2 at the target genes would have been informative as the authors start the manuscript in Figure 1, highlighting the changes in histone modifications which was very selective to only H3K36me2 and H3K27ac.

We thank the reviewer for this suggestion. As observed from our new ChIP data, changes in H3K36me2 are negatively correlated with differences in H3K27ac in LATS1-NCOR1 repressed regulatory regions (new Fig. S6E).

The authors mention utilizing their RNA-Seq data and mention that the NCOR1 activity has been compromised and it is not clear if it is based on pathway analysis of the genes, or the expression of NCOR1 itself. If it was pathway analysis, do they see the epigenetic modifier that contributes to H3K36me2 in the analysis. Motif analysis of their ATAC-seq would have given novel insights.

As mentioned above, decreased NCOR1 activity was inferred from IPA (QIAGEN) analysis of global gene expression patterns. Although LATS1 expression does not significantly affect NCOR1 mRNA levels (Fig. S6F), the expression of NCOR1 protein levels decreased in basal-like and in Lats1-CKO cultures (reviewer Fig. 1). Therefore, diminished NCOR1 activity might also be explained in part by decreased expression of NCOR1 protein, depending on LATS1 expression and cell state.

Reviewer Fig. 1: Lysates from WT and Lats1-CKO luminal and basal-like (BL) enriched cells were subjected to Western blot analysis and probed for NCOR1. GAPDH served as a loading control.

Unfortunately, there is no defined H3K36me2-specific motif. However, interestingly, upon reexamination of our IPA analysis, we noticed that both NSD2 and KDM2B (H3K36me2 methyl transferase and demethylase, respectively) appear as potential upstream regulators in two of the three datasets analyzed. Although beyond the scope of this paper, this brings up the intriguing possibility that regulation of H3K36me2 might also be important in determining epithelial cell state.

Reviewer #2 (Remarks to the Author):

In the present manuscript “Breast cancer plasticity is restricted by a LATS1-NCOR1 repressive axis” Aylon et al report that LATS1, helps maintain “a luminal identity” of PyMT mammary carcinoma cells by keeping “basal-specific” genes in a closed chromatin state through interaction with the nuclear corepressor NCOR1 and recruitment of HDAC1, driving histone H3K27 deacetylation.

The authors previously demonstrated using the MMTV-polyomavirus middle T oncogene (PyMT) driven mammary carcinoma model that deletion of LATS1 results in increased expression of “basal-like features” in the tumors with a reduced response to tamoxifen (Furth et al).

Here, they provide molecular underpinnings for this phenomenon in the PyMT-driven mammary carcinoma cells. Underlying the increased expression of basal features are epigenetic changes that involve opening of chromatin in regions encoding “basal genes”. The authors provide evidence that this phenomenon is YAP/TAZ independent. By showing that a Myc-tagged mouse LATS1 co-immunoprecipitates with endogenous NCOR1 in PyMT-carcinoma cells and that HDAC1 and NCOR1 only interact in the presence of LATS1 pointing to a ternary repressive complex that silences “basal genes”.

We would like to thank this reviewer for their insightful analysis of our manuscript.

Major concerns:

The data presented are clean and sound but the imprecise use of breast cancer-related terminology and the inadequacy of the models severely limit the biological interest of this work. The main claim about “breast cancer plasticity” is not supported by the data

To ensure the precise use of correct nomenclature, our pathologist critically re-reviewed the manuscript text. We color highlighted all our modifications and hope that now the text is less ambiguous.

To expand our findings into additional cancer models, we have replicated key findings in human breast cancer cell lines and included data from luminal breast cancer patients (detailed below).

To demonstrate the plasticity of cancer cell identity in our model, we employed both in vitro and in vivo strategies (former Figure 3A, B). Importantly, within a mammary tumor setting, luminal cells developed into tumors with markedly different histology depending on LATS1 status; whereas WT luminal cells generated poorly differentiated carcinomas, Lats1-CKO luminal cells generated basal-like sarcomatoid carcinomas (former Fig. S3B). Analogously, human breast cancer plasticity of luminal cell identity is clinically relevant. About half the patients with ER α -positive primary luminal breast tumors that relapse after adjuvant endocrine therapy have recurrent tumors in which luminal identity and ER α expression is lost⁴. Moreover, this inability to maintain luminal cell identity, which enables phenotypic plasticity and slippage into an ER-negative state⁵, is correlated with primary and acquired resistance to endocrine therapies⁶⁻⁸. These clinical observations illustrate the relevance of breast cancer plasticity, and particularly the ability to drift from luminal to basal-like cell state.

The authors assertion that decreased expression of LATS1 increases expression of basal genes and drives breast cancer cells (implying the human disease) into a basal program and tumor progression when the experimental evidence is merely based on the PyMT model. Ectopic

expression of this viral oncogene results in very fast growing tumors that bear little resemblance to the human disease.

The polyomavirus middle T oncogene (PyMT) induced mammary carcinomas express the estrogen receptor only in the early stages of tumorigenesis. Tumor growth has not been rigorously shown to be hormone-dependent; i.e. response to ovariectomy. Hence, the model reflects, if at all, human luminal B tumors only very initially. It is not clear at what age and stage of tumor development the experiments were performed.

We apologize for this oversight; mammary carcinomas from 3.5 month old PyMT mice were harvested for EpiTOF analysis and to generate PyMT tumor-derived cell lines. This has now been noted in the Results section (line 109).

Importantly, mammary lesions arising in the PyMT mice follow similar molecular and histological progression as human breast tumors⁹, making it a relevant tool for understanding breast cancer biology. The PyMT viral oncogene promotes cancer by constitutive activation of the same pathways (SRC, RAS, PI3K) that are hyperactivated by breast cancer-relevant oncogenes, such as receptor tyrosine kinases. Indeed, transformation by PyMT is strictly dependent on activation of PI3K^{10,11}. Notably, PIK3CA mutations are the most frequent oncogenic drivers in human luminal breast cancer. Consequently, the PyMT model has been used to study many aspects of breast cancer, including initiation¹², histological and molecular progression, metastasis^{10,13-21} and cancer immunology^{10,13-15,17,22}. Importantly, as cancer advances, progressive alterations lead to transcriptionally heterogeneous tumors, implying that full malignant transformation in this mouse model requires additional genetic and epigenetic events beyond PyMT expression^{18,23,24}. One such event is loss of estrogen signaling, similar to what occurs in endocrine-resistant Luminal B tumors²⁵⁻²⁷. The fact that similar aberrations are also detected in human breast cancer reinforces the view that tumor progression in this GEMM faithfully recapitulates the complex stages and heterogeneity of human breast cancer. Not surprisingly, the PyMT model has been a cornerstone for pre-clinical development and testing of potential therapies, and continues to be so, as evidenced also by publications from last year²⁸⁻³¹. Therefore, we believe that PyMT mice are an appropriate model for human luminal B breast cancer.

Despite these arguments in favor of the PyMT model, we agree with the reviewer with regard to the importance of demonstrating more formally the human relevance of our proposed molecular mechanism. In the previous version of our manuscript, we presented evidence that deletion of LATS1 resulted in an increase in H3K27ac also in human breast epithelial MCF10A cells (former Fig. 4C). In the current version, we made substantial efforts to demonstrate many of our key findings also in human luminal breast cancer-derived MCF7 and ZR751 cells. Briefly, we could confirm that, similar to PyMT cells, depletion of LATS1 (and/or NCOR1) in MCF7 and/or ZR751 cells also resulted in an increase in H3K27ac (new Fig. 4D) and upregulation of LATS1-NCOR1-repressed genes (new Fig. S5C), as well as in a global enrichment of ER-repressed gene expression (new Fig. 4H). Moreover, endogenous LATS1 and NCOR1 co-localization could be visualized in MCF7 cells (new Fig. S5F); the endogenous

proteins were co-immunoprecipitated (*new Fig. S5E*) and their interaction was confirmed to be mostly nuclear (*new Fig. S5I*).

Importantly, we have added experimental data from luminal breast cancer patient samples that support the notion of a functional interaction of *LATS1* and *NCOR1* also in human tumors (*new Fig. S7B*)(further details also below).

The terms “luminal” and “basal” designate the two major cell lineages in the normal mouse mammary epithelium and the normal human breast epithelium. The nomenclature for breast cancer subtypes is “luminal A or B” and “basal-like”. As the authors point out the latter term is misleading as basal-like breast carcinomas were shown to arise from luminal progenitor cells. Hence all breast carcinomas arise from luminal cells.

This imprecision in terminology confounds this manuscript. It is unclear what basal and luminal genes are: do the authors refer to the normal cell types in the breast, the mouse mammary epithelium, or are they related to molecular tumor subtypes? Moreover, these terms are intermingled with ER signatures.

As described (previous line 157), “luminal genes” are genes transcribed specifically in the FACS-enriched luminal subpopulation. We identified the luminal cell subpopulation by EpiTOF analysis of mouse mammary tumor epithelium, based on the relative abundance of EpCAM, CD49f, CD44 and CD24 cell surface markers, and the exclusion of CD45 and aSMA nonepithelial markers. We subsequently enriched luminal and basal-like cell populations based on the relative surface expression of EpCAM and CD49f.

*EpCAM and CD49f have been used previously to define cells of luminal and basal lineages from normal human breast tissue³²⁻³⁴. Notably, also primary breast cancer tumors have been categorized using EpCAM and CD49f to detect analogous luminal/basal-like subpopulations³⁵. In these abovementioned reports, there was a strong association between the cell surface-based categories and molecular markers of luminal cells (expressing *KRT8/18*) and basal cells (expressing *KRT14*). Importantly, the expression of an ER activated gene signature is fundamental to luminal cell identity³⁶.*

*We added a substantial amount of data to the current version of the manuscript to demonstrate the relevance of the luminal cells identified in this system to the histological luminal breast cancer subtype as well as the luminal molecular tumor subtype. Importantly, the gene expression pattern of the luminal mouse cell population was significantly similar to a human luminal breast cancer gene signature (*new Fig. 2E*), arguing that EpCAM and CD49f expression can serve as a proxy for luminal molecular subtype gene expression.*

*Additionally, we now present immunofluorescent staining of conventional luminal (*KRT8*) and basal-like markers (*KRT14*) in WT and *Lats1*-CKO luminal and basal-like cells (*new Fig. S2D*). As expected, *KRT8* was highly expressed in WT luminal cells and less so in the other cell populations. Likewise, *KRT14* was chiefly expressed in both WT and *Lats1*-CKO basal-like cells, with some positive cells observed also within the *Lats1*-CKO luminal population.*

*Moreover, mRNA levels of the conventional luminal markers *Krt8* and *Krt18* were higher in the EpCAM-CD49f enriched luminal subpopulation compared to the FACS-enriched basal-*

like subpopulation (*new Fig. S2B*). Likewise, the basal markers *Krt14* and *Krt5* were more highly expressed in the basal-like subpopulation; in fact, *Krt5* was solely expressed in our basal-like cells.

Minor comments:

FACS analysis: distinction of basal and luminal tumor cells is merely based on EpCAM expression levels. This is misleading when luminal tumors are ER+ and often progesterone receptor + while basal-like tumors lack hormone receptor expression. There is no reference to EpCAM expression levels distinguishing between tumor subtypes. CD49f not expressed in mature luminal and hormone receptor positive mammary epithelial cells but here it is high in both populations.

In addition to our comments above, we would like to draw the reviewer's attention to publications of EpCAM/CD49f FACS analyses (such as^{35,37}) in which both luminal and basal subpopulations were shown to harbor high levels of CD49f expression.

Figure 4E: the finding that LATS1 over expression increases levels of EpCAM detected on the cell surface is not sufficient to conclude on “luminal differentiation”.

In the previous version of our manuscript, we showed that overexpression of LATS1 in MDA-MB-468 cells is sufficient to shift these basal-like cells towards a more luminal-like phenotype as evaluated by FACS (as the reviewer pointed out, in previous Fig. 4E), gene expression patterns of specific LATS1-NCOR1 luminal-repressed targets (previous Fig. 5C) and instigation of a global luminal-like gene expression pattern of ER repressed genes (previous Fig. 2E) (now Figs. 4F, 5D and 2F, respectively).

*In the current version, we expanded our definition of luminal cells to include more established luminal cell markers (see comments above) and specifically interrogated the ability of LATS1 to drive a luminal phenotype. Towards this aim, we re-expressed LATS1 in Lats1-CKO cells and evaluated LATS1-dependent expression of the canonical luminal marker KRT8 (*new Fig. S4B*). We now show that expression of LATS1 significantly augmented the intensity and number of cells expressing KRT8, further strengthening the notion that LATS1 expression is sufficient to drive a transition towards a more luminal state.*

*Importantly, we compared the gene expression pattern of MDA-MB-468 cells following induced overexpression of LATS1 to the expression pattern of human luminal (compared to basal-like) breast cancers (*new Fig. 4G*). Here, also, we observed that LATS1 expression was sufficient to instigate a global luminal-like gene expression pattern in otherwise basal-like cells.*

Why chose MDA-MB-468 to test for the repressive effects of LATS1 overexpression?

Beyond the potential clinical relevance of shifting basal-like breast cancer cells towards a more luminal phenotype, it has been reported that MDA-MB-468 cells can undergo epigenetic reprogramming to transition to a luminal-like state^{38,39} (as also mentioned by us in the

Discussion). Nonetheless, we recognize the power of reproducing data in multiple model systems. For this reason, we made substantial efforts to demonstrate our key findings also in human luminal breast cancer cells (MCF7 and ZR751). Specifically, depletion of LATS1 from either MCF7 or ZR751 cells (described in⁴⁰) resulted in a global enrichment of ER repressed genes (new Fig. 4H), reinforcing the notion that LATS1 exerts repressive effects also in a luminal cell setting. To interrogate the role of NCOR1 in LATS1-dependent gene repression in MCF7 cells, we measured target gene expression following depletion of LATS1 with and without NCOR1 (new Fig. S5C). In all cases, gene expression increased as a result of decreased expression of LATS1 and/or NCOR1, suggesting that LATS1 represses NCOR1 targets also in human luminal breast cancer-derived cells. In line with the role of LATS1 and NCOR1 in transcriptional repression in MCF7 cells, depletion of either gene resulted in increased abundance of H3K27ac (new Fig. 4D).

To more fully elaborate the role of LATS1-NCOR1 in MCF7 cells, we validated the physical interaction of LATS1 and NCOR1 by endogenous Co-IP (new Fig. S5E) and examined their relative subcellular distribution pattern by immunofluorescence (new Fig. S5F). In all instances, the results using MCF7 cells were remarkably similar to those attained using the PyMT system.

Importantly, we added experimental data also from luminal breast cancer patient samples. Specifically, IHC staining of LATS1 and NCOR1 shows remarkably similar subcellular patterns of staining of both proteins, supporting the notion of a functional interaction of LATS1 and NCOR1 also in human tumors.

We believe that, together, our data confirms the existence of a LATS1-NCOR1 repressive axis also in a human luminal breast cancer setting.

Line 176: "both mouse and human cells" should read "both mouse mammary PyMT carcinoma cells and a human breast cancer cell line"

We have modified the statement as suggested (line 205).

The signature for "ER repressed" genes is based on a publication performed with microarrays in MCF7 cells in 2003 upon short time treatment of these cells in vitro.

We thank the reviewer for pointing this out to us. We have now reproduced our GSEA analyses using a more up-to-date ER repressed gene list⁴¹, with identical results (new Fig. S2G).

Figure 7B, labeling issue: x-axis.

We apologize for this oversight. We have now modified the x-axis labels of Fig. 7B to be clearer: "NCOR1^{low}LATS1^{low}" and "NCOR1^{high}LATS1^{high}".

Reviewer #3 (Remarks to the Author):

The manuscript by Aylon et al. studied the essential role of LATS1 in maintaining the luminal identity of breast tumor cells. The authors found that LATS1 interacted with NCOR1 and HDAC1 to regulate chromatin accessibility, which in turn, represses the expression of genes associated with the basal cell state. While this is an interesting paper, some conclusions were weakly supported by the data.

We thank the reviewer for their interest in our findings.

Major issues:

1. The data, in many cases, lack adequate biological replicates and lack sufficient statistical analysis. For example, the Western blotting, flow cytometry, and ATAC-seq only showed one sample for each group. More biologic replicates would strengthen the paper.

We thank the reviewer for this important point. We wish to emphasize that the key experiments all included biological replicates, as noted in the figure legends. Specifically, the EpiTOF data was based on three WT and three Lats1-CKO independently derived cell lines; the RNA-seq and ATAC-seq analyses were conducted on two biological replicates (cells derived from different mice) from each cell population; evaluation of cell identity plasticity (Fig. 3) is based on the average (and statistics) of 5 independent mouse mammary tumors; the FACS presented in Fig. 4A is based on three independent cell lines with reconstitution of MYC-LATS1; the RT-qPCR data presented in Fig. 5 is based on 3 biological repeats; and the ChIP experiments presented in Fig. 6 are based on 2 biological repeats.

During the revision process, we repeated many experiments to add even more biological replicates.

*Nonetheless, we agree with the reviewer that presenting data from all replicates is crucial for understanding the reproducibility of the results. In the present version of the paper, where relevant, we quantified FACS analyses and included graphical representations with individual data points of replicates. Additionally, ATAC-seq differential peak analysis was performed considering the variability between the two replicates. For coverage visualization (profiles), we merged the coverage files (BAM) from the replicates (except one L1-CKO luminal sample, in which a replicate was discarded due to unsatisfactory technical quality) (see **new Table S2**), so that now the average coverage from both replicates is shown. In all cases, we recalculated the statistics and modified/displayed p-values accordingly.*

2. Throughout the study, the authors define cell state by only two genes, CD49f and EpCAM. More genetic markers, a gene “signature”, are needed to provide more evidence of the basal or luminal cell state.

As indicated in the previous version, following FACS-based enrichment of luminal and basal-like populations on the basis of EpCAM and CD49f expression, global gene expression patterns

of these populations were compared. These luminal cells expressed genes enriched in the ERα activation gene signature when compared to basal-like cells (former Fig. 2D, now Fig. 2E, left), confirming their luminal identity.

In the revised version of this manuscript we added a substantial amount of data to further demonstrate the concordance of EpCAM+ luminal assignment with other definitions of luminal (or basal-like) cells. Importantly, the gene expression pattern of the EpCAM/CD49f-defined mouse luminal cancer cell subpopulation was significantly similar to that of a human luminal breast cancer gene signature (new Fig. 2E, right), strongly supporting the notion that EpCAM and CD49f expression is a satisfactory proxy for the luminal molecular subtype.

Additionally, we now present immunofluorescent staining using conventional luminal (KRT8) and basal-like markers (KRT14) in WT and Lats1-CKO luminal and basal-like cells (new Fig. S2D). As expected, KRT8 was more highly expressed in WT luminal cells and less so in the other cell populations. Likewise, KRT14 was chiefly expressed in both WT and Lats1-CKO basal-like cells, with some positive cells observed even within the Lats1-CKO luminal subpopulation.

We expanded this observation by comparing the mRNA levels and chromatin accessibility of two conventional luminal (Krt8 and Krt18) and two basal (Krt14 and Krt5) markers in WT luminal and basal-like cells (new Figs S2B, C). As expected, the luminal markers were more highly expressed in the luminal FACS-enriched subpopulation, whereas the basal markers were higher in the basal-like subpopulation, with Krt5 solely expressed in the basal-like cells.

3. In Fig 5, the similar trends of gene expression in LAST1-KO or NCOR1-KO cells may be correlation only. The authors should provide more evidence for their conclusion that LATS1 exerts its function through NCOR1. For example, checking gene expression in LATS1 overexpressed cells with NCOR1 deletion.

To address the reviewer's concern, we induced overexpression of LATS with or without knockdown of NCOR1, using the setup depicted in former Fig. 5C (now Fig. 5D). In all instances, knockdown of NCOR1 diminished the repressive effects of LATS1 overexpression (reviewer Fig. 2), providing more evidence that this function of LATS1 is dependent, at least in part, on NCOR1.

Reviewer Fig. 2: RT-qPCR analysis of NCOR1-repressed genes from MDA-MB-468 cells harboring vector control or inducible MYC-LATS1 (OE-LATS1). At the time of LATS1 induction, with 2uM doxycycline, cells were transfected with control siRNA against NCOR1 (siNCOR1). Cells were harvested 48 hours after LATS1 induction and NCOR1 depletion. Values were normalized to HPRT.

4. The mechanistic connection between LATS1 and NCOR1 is unclear. The key data regarding the interaction of LATS1 and NCOR1 is in an overexpression context and in vitro. Also, some controls are missing from the experiments.

We would like to apologize for not making the point clearer in our previous version; in Fig. S5D we presented co-immunoprecipitation of endogenous LATS1 and NCOR1 from PyMT cells (Fig. S5D). Of note, in the current version, we present also a reciprocal co-immunoprecipitation of endogenous LATS1 and endogenous NCOR1 from human MCF7 luminal breast cancer cells (new Fig. S5E). A rabbit antibody against His-tag served as an organism-relevant negative control in the immunoprecipitation.

Unfortunately, due to technical reasons (both the NCOR1 CST #5948 antibody and the LATS1 CST #3477 antibody are of rabbit origin) we were unable to perform proximity ligation assays (PLA) against endogenous mouse proteins, thus we had to resort to the use of GFP-LATS1 and a mouse anti-GFP antibody (Abcam, #ab1218). In repeated experiments (new Fig. S5H), no PLA signal was detected following depletion of endogenous NCOR1, which served as a negative control.

Happily, in a human cellular context, we were able to calibrate and apply a mouse-derived antibody against NCOR1 (#SC-515934). This was employed in PLA experiments in order to strengthen the data regarding the interaction of LATS1 and NCOR1 in an endogenous context, in MCF7 cells (new Fig. S5I). In this case, exclusion of a single primary antibody served as a negative control.

Importantly, we added experimental data also from luminal breast cancer patient samples. IHC staining of (endogenous and in vivo) LATS1 and NCOR1 showed remarkably similar subcellular patterns of expression, consistent with the notion of a functional interaction between LATS1 and NCOR1 also in human tumors.

We believe that, together, our data confirms the existence of a LATS1-NCOR1 repressive axis also in an endogenous human luminal breast cancer setting.

5. In Fig 5G, it appears that LATS1 and NCOR1 are not co-localized which argues against the two proteins forming a complex to regulate chromatin accessibility. This is also an overexpression system. Using a system with endogenous levels of proteins is more impactful. Moreover, quantification is required.

The reviewer raises an important concern. Yet, although most LATS1 is indeed cytoplasmic, some LATS1 can be detected also in the nucleus (Fig. 5G). Unfortunately, as mentioned above, both NCOR1 and LATS1 antibodies that are useful for staining of the mouse proteins are of rabbit origin, precluding double immunofluorescent staining of endogenous PyMT cell proteins. However, as we now show in new Fig. S5F, we could perform staining of endogenous LATS1 and NCOR1 in human luminal breast cancer MCF7 cells. Encouragingly, we could detect a partial overlap also in these cells, comparable to what we observed previously with the overexpressed tagged LATS1 in PyMT cells (Fig. 5G).

For the reason noted above, in the mouse setting we were obliged to use a tagged version of LATS1 in PLA experiments, which indicated that LATS1 and NCOR1 co-localize primarily (though not exclusively) in the nucleus (former supp Fig. 5E, now new Fig. S5H). To quantify colocalization, as requested by the reviewer, and to exclude the possibility that the large GFP-tag might affect the proper subcellular distribution of LATS1, we have now performed ImageStream analysis using a MYC-tagged version of mouse LATS1. In these experiments, we found that 15% of LATS1 and 93% of NCOR1 were nuclear. Remarkably, by this approach, 80% of cells revealed a significant overlap of LATS1 and NCOR1 distribution (Similarity > 1.5, Z-Score = 0.013). This data is now presented in new Fig. S5G. Notably, the ImageStream analysis also enabled quantification of our second observation from Fig. 5G, namely that LATS1 positively affects NCOR1 levels (former lines 267-270). Indeed, endogenous NCOR1 staining intensity significantly decreased in PyMT cells with knockout of endogenous Lats1, compared to WT cells. Moreover, following overexpression of LATS1, endogenous NCOR1 intensity significantly increased. Graphical representation of this data is presented in the new Fig. S5G, panel 4.

Minor issues:

1. Fig 1A, it would be better to show a bar graph indicating the percentages of different cell populations.

We thank the reviewer for this helpful suggestion. We have now incorporated a bar graph indicating the percentages of different cell subpopulations for each of the EpiTOF samples (new Fig. S1D, right panel).

2. Fig 1C is confusing.

We apologize for the confusion and agree with the reviewer. We now present the raw data as violin plots in a new Fig. 1C.

3. Fig 1E, the expression of CD49f seems to be similar in all three cell types. Please clarify.

We agree with the reviewer that there are only very mild differences in CD49f levels between the three subpopulations, which are mostly separated by differential EpCAM expression. However, we are not unique in this observation and would like to draw the attention of the reviewer to previously published EpCAM/CD49f FACS analyses (e.g.^{35,37}) in which both luminal and basal subpopulations harbor high levels of CD49f expression.

4. Fig 2A and 2B, please clarify the method for normalizing cell numbers for the sequencing experiment.

We apologize for this unintentional oversight on our part. We now include in the text (line 143) and legends of Fig. 2A, B that cells were briefly expanded to attain a minimum of 50,000 cells for analysis.

5. Fig 2C bottom, the read counts of Basal genes were lower in WT basal cells as compared to WT luminal cells. Please clarify this

Like the reviewer, we also expected chromatin surrounding genes upregulated in basal-like cells to be more accessible in WT basal-like cells. When looking at our new figure that presents merged coverage from both ATAC-seq repeats (thanks to this reviewer’s suggestion in relation to repeats), this discrepancy still exists, albeit milder. We believe that there is no significant difference between Lats1-CKO luminal, WT luminal and WT basal-like accessibility in these regions, suggesting that although chromatin accessibility is necessary, other factors influence the extent of “basal” gene expression.

In line with this notion, we now show that enhancers associated with genes more highly expressed in basal-like cells were more accessible in WT basal-like compared to WT luminal cells (new Fig. 2D, right). Interestingly, enhancers associated with genes more highly expressed in luminal cells displayed no differential accessibility in any of the four cell types (new Fig. 2D, left), suggesting that LATS1-NCOR1 might function to keep aberrant “basal” enhancers inaccessible in luminal cells. The text has been rewritten accordingly, to accommodate this additional information.

6. Fig 2E bottom. It might be better to represent the data as a heat map with representative genes.

As suggested by the reviewer, below is a heat map portraying the expression of ER-repressed genes from the leading edge of the GSEA performed in former Fig. 2E (now Fig. 2F, right), depicting three biological repeats (reviewer Fig. 3). As expected, overall, ER repressed genes are expressed at lower levels following induction of LATS1 expression.

Reviewer Fig. 3: The expression of genes contributing to the leading edge of the GSEA analysis (presented in Fig. 2F, right). Gene expression was extracted from RNA-seq data of vector vs overexpression of LATS1 (OE-LATS1). Three biological replicates are displayed with color representing row Z-score of expression (red high and blue low).

7. Fig 3A, please show the percentages of each population. Is it that possible the luminal cells grow faster than basal cells in culture conditions?

As requested, we have added the percentages of each population to all FACS figures in the manuscript.

As suggested by the reviewer, we compared the cell cycle distribution patterns of the cultured luminal and basal-like cells. In fact, this BrDU analysis demonstrated that basal-like cells had a larger S phase fraction than luminal cells (reviewer Fig. 4), arguing against the possibility that within the experimental conditions of Fig. 3A, luminal cells are outgrowing the basal-like subpopulation.

Reviewer Fig. 4: WT luminal and WT basal-like enriched cells were propagated in standard culturing conditions. Cells were pulsed for 45 minutes with 10uM BrdU, harvested and incubated with an anti-BrdU-FITC conjugated antibody. Cell cycle distribution was based on incorporation of propidium iodide (upper panels). The red box labeled S (lower panels) represents the portion of cells undergoing proliferation. No BrdU pulse served as a negative control.

We thank the reviewer for this suggestion. We performed Western blot analysis to evaluate the level of phosphorylated YAP/TAZ (reviewer Fig. 5). As expected, amounts of phosphorylated YAP increase in *Lats1-CKO* PyMT cells. This is concurrent with an increase in total YAP protein levels. Interestingly, this trend is also seen in WT basal-like cells. Together with the YAP/TAZ target activation in Fig. S4D, this suggests that YAP is more active in *Lats1-CKO* cells than in WT PyMT cells. Unfortunately, our antibody (sc-17610) against the phosphorylated form of TAZ did not detect mouse phospho-TAZ (despite indications in the datasheet). Nonetheless, since depletion of either YAP or TAZ had no effect on the fraction of basal-like cells (now supp Fig. S4E, F), it still seems strongly unlikely that LATS1-driven promotion of luminal phenotypes occurs via inhibition of YAP or TAZ in this model.

Reviewer Fig. 5: WT and Lats1-CKO luminal and basal-like enriched cells were subjected to Western blot analysis with the indicated antibodies. GAPDH served as the loading control.

9. Please clarify the cell types used in Fig 6A and 6B?

We apologize for this oversight. The cells used in former Fig. 6A, B were WT and Lats1-CKO PyMT cells. This has now been noted in the figure legend.

References

- 1 Fang, Y. *et al.* The H3K36me2 methyltransferase NSD1 modulates H3K27ac at active enhancers to safeguard gene expression. *Nucleic Acids Res* **49**, 6281-6295, doi:10.1093/nar/gkab473 (2021).
- 2 Chen, H. *et al.* H3K36 dimethylation shapes the epigenetic interaction landscape by directing repressive chromatin modifications in embryonic stem cells. *Genome Res* **32**, 825-837, doi:10.1101/gr.276383.121 (2022).
- 3 Lhoumaud, P. *et al.* NSD2 overexpression drives clustered chromatin and transcriptional changes in a subset of insulated domains. *Nat Commun* **10**, 4843, doi:10.1038/s41467-019-12811-4 (2019).
- 4 Johnston, S. R. *et al.* Changes in estrogen receptor, progesterone receptor, and pS2 expression in tamoxifen-resistant human breast cancer. *Cancer Res* **55**, 3331-3338 (1995).
- 5 Nakao, M., Fujiwara, S. & Iwase, H. Cancer Navigation Strategy for Endocrine Therapy-Resistant Breast Tumors. *Trends Cancer* **4**, 404-407, doi:10.1016/j.trecan.2018.04.005 (2018).
- 6 Haque, R. *et al.* Impact of breast cancer subtypes and treatment on survival: an analysis spanning two decades. *Cancer Epidemiol Biomarkers Prev* **21**, 1848-1855, doi:10.1158/1055-9965.EPI-12-0474 (2012).
- 7 Ignatiadis, M. & Sotiriou, C. Luminal breast cancer: from biology to treatment. *Nat Rev Clin Oncol* **10**, 494-506, doi:10.1038/nrclinonc.2013.124 (2013).
- 8 Jeselsohn, R., Buchwalter, G., De Angelis, C., Brown, M. & Schiff, R. ESR1 mutations—a mechanism for acquired endocrine resistance in breast cancer. *Nat Rev Clin Oncol* **12**, 573-583, doi:10.1038/nrclinonc.2015.117 (2015).
- 9 Attalla, S., Taifour, T., Bui, T. & Muller, W. Insights from transgenic mouse models of PyMT-induced breast cancer: recapitulating human breast cancer progression in vivo. *Oncogene* **40**, 475-491, doi:10.1038/s41388-020-01560-0 (2021).
- 10 Lin, E. Y. *et al.* Progression to malignancy in the polyoma middle T oncoprotein mouse breast cancer model provides a reliable model for human diseases. *Am J Pathol* **163**, 2113-2126, doi:10.1016/S0002-9440(10)63568-7 (2003).
- 11 Pylayeva, Y. *et al.* Ras- and PI3K-dependent breast tumorigenesis in mice and humans requires focal adhesion kinase signaling. *J Clin Invest* **119**, 252-266, doi:10.1172/JCI37160 (2009).
- 12 Decock, J. *et al.* Pleiotropic functions of the tumor- and metastasis-suppressing matrix metalloproteinase-8 in mammary cancer in MMTV-PyMT transgenic mice. *Breast Cancer Res* **17**, 38, doi:10.1186/s13058-015-0545-8 (2015).
- 13 Wyckoff, J. B. *et al.* Direct visualization of macrophage-assisted tumor cell intravasation in mammary tumors. *Cancer Res* **67**, 2649-2656, doi:10.1158/0008-5472.CAN-06-1823 (2007).
- 14 DeNardo, D. G. *et al.* CD4(+) T cells regulate pulmonary metastasis of mammary carcinomas by enhancing protumor properties of macrophages. *Cancer Cell* **16**, 91-102, doi:10.1016/j.ccr.2009.06.018 (2009).
- 15 Lin, E. Y., Nguyen, A. V., Russell, R. G. & Pollard, J. W. Colony-stimulating factor 1 promotes progression of mammary tumors to malignancy. *J Exp Med* **193**, 727-740, doi:10.1084/jem.193.6.727 (2001).
- 16 Maglione, J. E. *et al.* Transgenic Polyoma middle-T mice model premalignant mammary disease. *Cancer Res* **61**, 8298-8305 (2001).

- 17 Jones, L. M. *et al.* STAT3 Establishes an Immunosuppressive Microenvironment during the Early Stages of Breast Carcinogenesis to Promote Tumor Growth and Metastasis. *Cancer Res* **76**, 1416-1428, doi:10.1158/0008-5472.CAN-15-2770 (2016).
- 18 Hirukawa, A. *et al.* Targeting EZH2 reactivates a breast cancer subtype-specific anti-metastatic transcriptional program. *Nat Commun* **9**, 2547, doi:10.1038/s41467-018-04864-8 (2018).
- 19 Schwab, L. P. *et al.* Hypoxia-inducible factor 1alpha promotes primary tumor growth and tumor-initiating cell activity in breast cancer. *Breast Cancer Res* **14**, R6, doi:10.1186/bcr3087 (2012).
- 20 Cook, R. S. *et al.* ErbB3 ablation impairs PI3K/Akt-dependent mammary tumorigenesis. *Cancer Res* **71**, 3941-3951, doi:10.1158/0008-5472.CAN-10-3775 (2011).
- 21 Lukes, L., Crawford, N. P., Walker, R. & Hunter, K. W. The origins of breast cancer prognostic gene expression profiles. *Cancer Res* **69**, 310-318, doi:10.1158/0008-5472.CAN-08-3520 (2009).
- 22 Gross, E. T. *et al.* Immunosurveillance and immunoediting in MMTV-PyMT-induced mammary oncogenesis. *Oncoimmunology* **6**, e1268310, doi:10.1080/2162402X.2016.1268310 (2017).
- 23 Cai, Y. *et al.* Transcriptomic dynamics of breast cancer progression in the MMTV-PyMT mouse model. *BMC Genomics* **18**, 185, doi:10.1186/s12864-017-3563-3 (2017).
- 24 Cai, Y. *et al.* Epigenetic alterations to Polycomb targets precede malignant transition in a mouse model of breast cancer. *Sci Rep* **8**, 5535, doi:10.1038/s41598-018-24005-x (2018).
- 25 Szostakowska, M., Trebinska-Stryjewska, A., Grzybowska, E. A. & Fabisiwicz, A. Resistance to endocrine therapy in breast cancer: molecular mechanisms and future goals. *Breast Cancer Res Treat* **173**, 489-497, doi:10.1007/s10549-018-5023-4 (2019).
- 26 Kittaneh, M., Montero, A. J. & Gluck, S. Molecular profiling for breast cancer: a comprehensive review. *Biomark Cancer* **5**, 61-70, doi:10.4137/BIC.S9455 (2013).
- 27 Osborne, C. K. & Schiff, R. Mechanisms of endocrine resistance in breast cancer. *Annu Rev Med* **62**, 233-247, doi:10.1146/annurev-med-070909-182917 (2011).
- 28 Lal, J. C. *et al.* Comparing syngeneic and autochthonous models of breast cancer to identify tumor immune components that correlate with response to immunotherapy in breast cancer. *Breast Cancer Res* **23**, 83, doi:10.1186/s13058-021-01448-1 (2021).
- 29 Luond, F. *et al.* Distinct contributions of partial and full EMT to breast cancer malignancy. *Dev Cell* **56**, 3203-3221 e3211, doi:10.1016/j.devcel.2021.11.006 (2021).
- 30 Li, L., Wang, J., Radford, D. C., Kopecek, J. & Yang, J. Combination treatment with immunogenic and anti-PD-L1 polymer-drug conjugates of advanced tumors in a transgenic MMTV-PyMT mouse model of breast cancer. *J Control Release* **332**, 652-659, doi:10.1016/j.jconrel.2021.02.011 (2021).
- 31 Vogel, C. F. A. *et al.* Targeting the Aryl Hydrocarbon Receptor Signaling Pathway in Breast Cancer Development. *Front Immunol* **12**, 625346, doi:10.3389/fimmu.2021.625346 (2021).
- 32 Villadsen, R. *et al.* Evidence for a stem cell hierarchy in the adult human breast. *J Cell Biol* **177**, 87-101, doi:10.1083/jcb.200611114 (2007).

- 33 Lim, E. *et al.* Aberrant luminal progenitors as the candidate target population for basal tumor development in BRCA1 mutation carriers. *Nat Med* **15**, 907-913, doi:10.1038/nm.2000 (2009).
- 34 Eirew, P. *et al.* A method for quantifying normal human mammary epithelial stem cells with in vivo regenerative ability. *Nat Med* **14**, 1384-1389, doi:10.1038/nm.1791 (2008).
- 35 Keller, P. J. *et al.* Mapping the cellular and molecular heterogeneity of normal and malignant breast tissues and cultured cell lines. *Breast Cancer Res* **12**, R87, doi:10.1186/bcr2755 (2010).
- 36 Bertucci, F. *et al.* How different are luminal A and basal breast cancers? *Int J Cancer* **124**, 1338-1348, doi:10.1002/ijc.24055 (2009).
- 37 Prat, A. *et al.* Characterization of cell lines derived from breast cancers and normal mammary tissues for the study of the intrinsic molecular subtypes. *Breast Cancer Res Treat* **142**, 237-255, doi:10.1007/s10549-013-2743-3 (2013).
- 38 Bussard, K. M. & Smith, G. H. Human breast cancer cells are redirected to mammary epithelial cells upon interaction with the regenerating mammary gland microenvironment in-vivo. *PLoS One* **7**, e49221, doi:10.1371/journal.pone.0049221 (2012).
- 39 Arabsolghar, R., Azimi, T. & Rasti, M. Mutant p53 binds to estrogen receptor negative promoter via DNMT1 and HDAC1 in MDA-MB-468 breast cancer cells. *Mol Biol Rep* **40**, 2617-2625, doi:10.1007/s11033-012-2348-7 (2013).
- 40 Furth, N. *et al.* LATS1 and LATS2 suppress breast cancer progression by maintaining cell identity and metabolic state. *Life Sci Alliance* **1**, e201800171, doi:10.26508/lisa.201800171 (2018).
- 41 Cicatiello, L. *et al.* Estrogen receptor alpha controls a gene network in luminal-like breast cancer cells comprising multiple transcription factors and microRNAs. *Am J Pathol* **176**, 2113-2130, doi:10.2353/ajpath.2010.090837 (2010).

REVIEWERS' COMMENTS

Reviewer #1 (Remarks to the Author):

Aylon et al have responded to their best to my comments. The work is significant and an important contribution to the field. I recommend for publication pending editor's decision

Reviewer #2 (Remarks to the Author):

The reviewer appreciates the efforts made by the authors and the minor concerns were addressed. The major concerns remain, the authors make a claim about luminal breast cancer plasticity based on unconnected findings in various models of questionable relevance to the disease.

Regrettably, the terms "basal-like" and "luminal" introduced to label different molecular subtypes have led to a lot of confusion in the breast cancer field and may also confound the author. The following article by Barry Gusterson might be helpful.

<https://breast-cancer-research.biomedcentral.com/articles/10.1186/bcr1041>

For a GEMM to be considered a good model for ER+ breast cancer, the following two criteria should be matched.

1. There should be data to support that these tumors are strongly ER+
2. The tumors should be hormone sensitive, that is they should growth arrest upon ovariectomy.

To my knowledge neither has been shown for MMTV PyMT tumors as used in this study.

The authors now provide the information that the data shown in Figure 1A at the heart of this manuscript are derived from 3.5-month-old mice. At this stage, palpable tumors are present in the mammary glands of MMTV PyMT transgenic females.

According to the characterization by Lin et al [https://doi.org/10.1016/S0002-9440\(10\)63568-7](https://doi.org/10.1016/S0002-9440(10)63568-7) there is no ER expression left at this point.

The authors argue that as ER+ tumors progress they lose their luminal properties and ER expression. This does not reflect clinical observations. ER expression is preserved through to metastatic progression. The clinical problem is endocrine resistance that arises with adjuvant therapy, resulting typically in ligand-independent ER signaling, constitutively active ER signaling this may or may not entail down modulation of ER expression at protein level.

As such there is only in vitro transient transfection data of two ER+ breast cancer cell lines showing a decrease in ER transcriptional signatures in response to LATS down modulation. This is not sufficient to propose that Lats modulation increases luminal BC cell lineage plasticity. Hence the clinical implications of this work remain unclear.

Reviewer #3 (Remarks to the Author):

concerns have been adequately addressed. The authors did a considerable amount of extra work to address the weaknesses and the paper is now improved

Reviewer #2

The reviewer appreciates the efforts made by the authors and the minor concerns were addressed. The major concerns remain, the authors make a claim about luminal breast cancer plasticity based on unconnected findings in various models of questionable relevance to the disease.

Regrettably, the terms “basal-like” and “luminal” introduced to label different molecular subtypes have lead to a lot of confusion in the breast cancer field and may also confound the author. The following article by Barry Gustersson might be helpful.

<https://breast-cancer-research.biomedcentral.com/articles/10.1186/bcr1041>

We thank the reviewer for pointing out this important review, highlighting the complexity of defining cellular identities within mammary tumors. While the terms luminal A, luminal B (hereafter LumA and LumB) and basal-like have been introduced in an attempt to classify different molecular subtypes of breast cancer, they are obviously a gross oversimplification. With the advent of more comprehensive molecular profiling methods, it has become clear that these are not distinct entities but rather a spectrum of cell states, with substantial inter-tumor and intra-tumor diversity. This further exemplifies that a sufficiently comprehensive set of molecular markers is needed to define phenotypic heterogeneity within a cancer cell population. This is one of the reasons why we started our study with a multi-parametric approach (epiTOF). Furthermore, we now cite Gusterson et al. when introducing the use of keratins to distinguish between cellular phenotypic identities.

We also agree with the reviewer that strict adherence to terminologies can sometimes be confusing, particularly if different researchers do not always have the same entity in mind. Therefore, in the previous revision we changed “basal” to “basal-like”. We have now further modified it, when appropriate, to say that LATS1 depletion favors a “more basal-like” state. We feel that this definition is broad enough to encompass the different views and is fully supported by our experimental data.

For a GEMM to be considered a good model for ER+ breast cancer, the following two criteria should be matched.

- 1. There should be data to support that these tumors are strongly ER+*
 - 2. The tumors should be hormone sensitive, that is they should growth arrest upon ovariectomy.*
- To my knowledge neither has been shown for MMTV PyMT tumors as used in this study.*

The authors now provide the information that the data shown in Figure 1A at the heart of this manuscript are derived from 3.5-month-old mice. At this stage, palpable tumors are present in the mammary glands of MMTV PyMT transgenic females.

According to the characterization by Lin et al [https://doi.org/10.1016/S0002-9440\(10\)63568-7](https://doi.org/10.1016/S0002-9440(10)63568-7) there is no ER expression left at this point.

The authors argue that as ER+ tumors progress they lose their luminal properties and ER expression. This does not reflect clinical observations. ER expression is preserved through to

metastatic progression. The clinical problem is endocrine resistance that arises with adjuvant therapy, resulting typically in ligand-independent ER signaling, constitutively active ER signaling this may or may not entail down modulation of ER expression at protein level.

We agree with the reviewer that the MMTV-PyMT model has its drawbacks. No breast cancer GEMM is a perfect reproduction of human breast cancer. This is due not only to the differences in mouse vs human mammary biology and physiology, but also largely to the fact that human breast tumors are often much more heterogeneous and complex than the relatively simple genetically engineered mouse tumors. Yet, we believe that the MMTV-PyMT model captures many of the features of human luminal B breast cancer, as discussed comprehensively in the 2021 review by Attalla et al. In particular, it recapitulates aspects of lumB progression, which is the key theme of our study. Furthermore, although progressive loss of progesterone receptor is more common, acquired resistance to endocrine therapy is sometimes also associated with substantial downregulation of ER expression (Attalla et al. 2021). Importantly, this has been reported to occur preferentially in LumB but not LumA tumors (Ahn et al. 2013; Osborne and Schiff, 2011; Szostakowska et al. 2019). We also note that the revised manuscript includes several experiments, employing human breast cancer cell lines of both luminal and basal-like origin, that support the results obtained with the MMTV-PyMT mouse model.

Nevertheless, to accommodate the reviewer's concerns, we have now modified the text to acknowledge the limitations of the PyMT mouse model.

As such there is only in vitro transient transfection data of two ER+ breast cancer cell lines showing a decrease in ER transcriptional signatures in response to LATS down modulation. This is not sufficient to propose that Lats modulation increases luminal BC cell lineage plasticity. Hence the clinical implications of this work remain unclear.

With regard to the clinical relevance of our study, we would like to point out that our observations from the mouse model are supported by the analysis of expression data derived from human breast cancer tumors, which is included in the paper.

We also note that LumB tumors display substantial intratumoral phenotypic diversity, including a variable fraction of basal-like cells (Georgopoulou et al.). Hence, studying factors that encourage a shift of LumB cells towards a more basal-like phenotype is of clear clinical relevance. Lastly, we would like to reiterate that, as reported previously, significantly reduced expression of LATS1 is a distinct feature of human LumB tumors (see Fig. 1C in Furth et al. 2018), providing a compelling rationale for investigating the impact of LATS1 loss particularly in the context of LumB breast cancer.

In view of the above, and in an attempt to accommodate the reviewer's concerns, we have now tried to explain more clearly that we are modeling LumB cancer, rather than ER+ cancer in general.

We hope that you will find these explanations satisfactory and will approve the publication of our MS in Nature Communications.

References:

Gusterson, B. A., Ross, D. T., Heath, V. J. & Stein, T. Basal cytokeratins and their relationship to the cellular origin and functional classification of breast cancer. *Breast Cancer Res* 7, 143, doi:10.1186/bcr1041 (2005).

Attalla, S., Taifour, T., Bui, T. & Muller, W. Insights from transgenic mouse models of PyMT-induced breast cancer: recapitulating human breast cancer progression in vivo. *Oncogene* 40, 475-491, doi:10.1038/s41388-020-01560-0 (2021).

Ahn, H. J., Jung, S. J., Kim, T. H., Oh, M. K. & Yoon, H. K. Differences in Clinical Outcomes between Luminal A and B Type Breast Cancers according to the St. Gallen Consensus 2013. *Journal of breast cancer* 18, 149-159, doi:10.4048/jbc.2015.18.2.149 (2015).

Osborne, C. K. & Schiff, R. Mechanisms of endocrine resistance in breast cancer. *Annual review of medicine* 62, 233-247, doi:10.1146/annurev-med-070909-182917 (2011).

Szostakowska, M., Trębińska-Stryjewska, A., Grzybowska, E. A. & Fabisiwicz, A. Resistance to endocrine therapy in breast cancer: molecular mechanisms and future goals. *Breast cancer research and treatment* 173, 489-497, doi:10.1007/s10549-018-5023-4 (2019).

Georgopoulou, D. *et al.* Landscapes of cellular phenotypic diversity in breast cancer xenografts and their impact on drug response. *Nat Commun* 12, 1998, doi:10.1038/s41467-021-22303-z (2021).

Furth, N. *et al.* LATS1 and LATS2 suppress breast cancer progression by maintaining cell identity and metabolic state. *Life Sci. Alliance* 1, e201800171, doi:10.26508/lsa.201800171 (2018).